# The temporal and spectral characteristics of expectations and prediction errors in pain and thermoception

**Andreas Strube\*, Michael Rose, Sepideh Fazeli, Christian Büchel\***

Department of Systems Neuroscience, University Medical Center Hamburg-Eppendorf, Hamburg, Germany

**Abstract** In the context of a generative model, such as predictive coding, pain and heat perception can be construed as the integration of expectation and input with their difference denoted as a prediction error. In a previous neuroimaging study (Geuter et al., 2017) we observed an important role of the insula in such a model but could not establish its temporal aspects. Here, we employed electroencephalography to investigate neural representations of predictions and prediction errors in heat and pain processing. Our data show that alpha-to-beta activity was associated with stimulus intensity expectation, followed by a negative modulation of gamma band activity by absolute prediction errors. This is in contrast to prediction errors in visual and auditory perception, which are associated with increased gamma band activity, but is in agreement with observations in working memory and word matching, which show gamma band activity for correct, rather than violated, predictions.

## Introduction

It has been shown that physically identical nociceptive input can lead to variable sensations of pain, depending on contextual factors (*Tracey and Mantyh, 2007*). In particular, attention, reappraisal, and expectation are core mechanisms that influence how nociception leads to pain (*Wiech et al., 2008*). A clinically important example of how expectations can shape pain processing is placebo hypoalgesia: pain relief mediated by expectation and experience – in the absence of active treatment (*Petrovic et al., 2002*; *Wager et al., 2004*; *Colloca and Benedetti, 2005*; *Bingel et al., 2006*; *Atlas and Wager, 2012*; *Anchisi and Zanon, 2015*).

In the context of a generative model of pain, it has been proposed that pain perception can be seen as the consequence of an integration of expectations with nociception (*Büchel et al., 2014*; *Wiech, 2016*; *Ongaro and Kaptchuk, 2019*). In this framework, expectations are integrated with incoming nociceptive information and both are weighted by their relative precision (*Grahl et al., 2018*) to form a pain percept. This can be seen in analogy to ideas in multisensory integration (*Ernst and Banks, 2002*). Expectations or predictions and resulting prediction errors also play a key role in generative models such as predictive coding (*Huang et al., 2011*). In essence, this framework assumes that neuronal assemblies implement perception and learning by constantly matching incoming sensory data with the top-down predictions of an internal or generative model (*Knill and Pouget, 2004*; *Huang et al., 2011*; *Clark, 2013*). Basically, minimizing prediction errors allows systems to resist their tendency to disorder by the creation of models with better predictions regarding the sensory environment, leading to a more efficient encoding of information (*Friston, 2010*).

Electroencephalogram (EEG) correlates of nociceptive skin stimulation have been widely investigated. Generally, phasic gamma activity has been associated with stimulus intensity over the sensory cortex where the amplitudes of pain-induced gamma oscillations increase with objective stimulus intensity and subjective pain intensity (*Gross et al., 2007*; *Hauck et al., 2007*; *Zhang et al., 2012*;

**\*For correspondence:**
a.strube@uke.de (AS);
buechel@uke.de (CB)

**Competing interests:** The authors declare that no competing interests exist.

*Rossiter et al., 2013*; *Tiemann et al., 2015*). Additionally, pain-related gamma band oscillations have been linked to the insular cortex as well as temporal and frontal regions using depth electrodes in epilepsy patients (*Liberati et al., 2018*). In tonic painful heat stimulation, medial prefrontal gamma activity has been observed (*Schulz et al., 2015*). In addition, gamma activity is enhanced by attention in human EEG experiments in visual (*Gruber et al., 1999*), auditory (*Tiitinen et al., 1993*; *Debener et al., 2003*), and sensorimotor processing (i.e. tactile stimuli) (*Bauer et al., 2006*) as well as in nociception (*Hauck et al., 2007*; *Hauck et al., 2015*; *Tiemann et al., 2010*).

Pain-related alpha-to-beta band oscillations are typically found to be suppressed with higher stimulus intensity (*Mouraux et al., 2003*; *Ploner et al., 2006*; *May et al., 2012*; *Hu et al., 2013*), which is enhanced by attention (*May et al., 2012*) and (placebo) expectation (*Huneke et al., 2013*; *Tiemann et al., 2015*; *Albu and Meagher, 2016*). Interestingly, prestimulus theta (*Taesler and Rose, 2016*) as well as prestimulus alpha and gamma activity (*Tu et al., 2016*) can affect subsequent pain processing. Specifically, trials with smaller prestimulus alpha and gamma oscillations were perceived as more painful, suggesting a negative modulation of subsequent pain perception (*Tu et al., 2016*).

Cued pain paradigms (*Atlas et al., 2010*) have been used to generate expectations and prediction errors. Previous functional magnetic resonance imaging (fMRI) studies have suggested an important role of the anterior insular cortex for mediating expectation effects and the integration of prior expectation and prediction errors in the context of pain (*Ploghaus et al., 1999*; *Koyama et al., 2005*; *Atlas et al., 2010*; *Geuter et al., 2017*; *Fazeli and Büchel, 2018*). These studies have revealed that neuronal signals in the anterior insula represent predictions and prediction errors with respect to pain, which in theory would allow the combination of both terms as required for predictive coding (*Büchel et al., 2014*; *Ongaro and Kaptchuk, 2019*). However, in fMRI studies, predictions and prediction errors cannot be temporally dissociated due to the low temporal resolution of the method. To investigate this further, we conducted a cue-based pain experiment using EEG to achieve high temporal and spectral resolution of predictions and prediction error processes in the context of pain.

In this experiment (N = 29) we employed contact heat stimuli with three different intensities (low heat, medium heat, and high heat), preceded by a visual cue indicating the upcoming intensity (*Figure 1*). To generate prediction errors, the modality (picture or heat) was correctly cued only in 70% of all trials, and stimulus intensities were correctly cued only in 60% of all trials. We then investigated oscillatory activity related to stimulus intensity, expectation, and prediction errors (*Figure 2*).

Based on the previous data, we hypothesized that expectation signals should temporally precede prediction error signals. Based on the functional neuroanatomy of cortical microcircuits (*Bastos et al., 2012*), with feedforward connections predominately originating from superficial layers and feedback connections from deep layers, we expect that prediction error signals should be related to higher frequencies (e.g. gamma band) than prediction signals (*Todorovic et al., 2011*; *Arnal and Giraud, 2012*).

## Materials and methods

### Participants

We investigated 35 healthy male participants (mean 26, range 18–37 years), who were paid as compensation for their participation. Applicants were excluded if one of the following exclusion criteria applied: neurological, psychiatric, dermatological diseases, pain conditions, current medication, or substance abuse. All volunteers gave their informed consent. The study was approved by the Ethics Board of the Hamburg Medical Association. Of 35 participants, data from six participants had to be excluded from the final EEG data analysis due to technical issues during the EEG recording (i.e. the data of the excluded participants were contaminated with excessive muscle and/or technical artifacts) leaving a final sample of 29 participants. The sample size was determined according to a power calculation (G*Power V 3.1.9.4) based on *Geuter et al., 2017*. For the left anterior insula (fMRI; Table 1 in *Geuter et al., 2017*), we observed an effect size of partial eta squared of 0.17 and an effect size of 0.22 for the right anterior insula (cue × stimulus interaction). Using a power of (1-

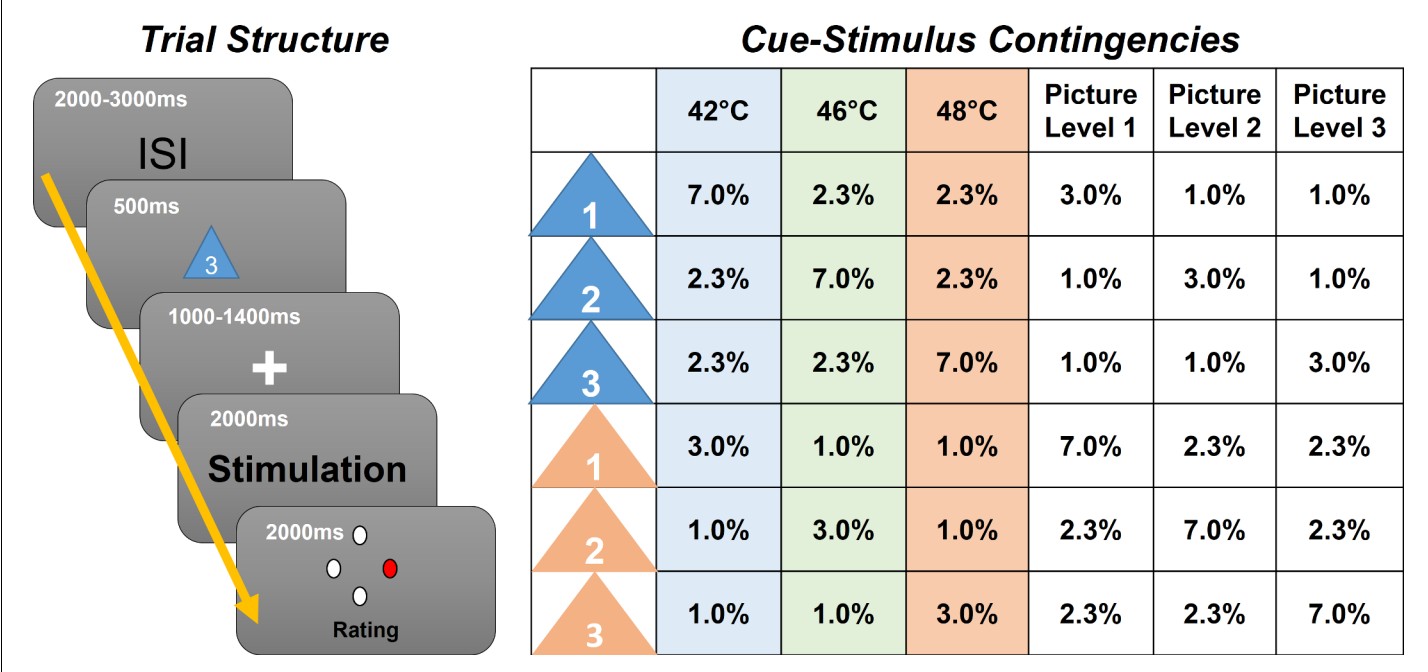

**Figure 1.** Left: Graphical representation of the trial structure. Each trial started with the presentation of a cue, indicating the stimulus intensity and modality of the following stimulus. After a jittered phase where only the fixation cross was shown, the stimulus (visual or thermal) was presented. A rating phase (1–4) of the stimulus aversiveness followed. Right: Contingency table for all conditions for each cue–stimulus combination. Note that percentages are for all trials; therefore, each row adds up to 1/6 (six different cues).

The online version of this article includes the following figure supplement(s) for figure 1:

**Figure supplement 1.** Histogram showing the distribution of the total number of rejected components based on detected muscle artifacts.

beta) of 0.95 and an alpha level of 0.05 and assuming a low correlation (0.1) between repeated measures, this leads to a sample size of 25, assuming the weaker effect in the left insula.

## Stimuli and task

Stimulus properties were chosen to be identical to a previous fMRI study of predictive coding in pain where both expectation and absolute prediction error effects were observed (*Fazeli and Büchel, 2018*). Thermal stimulation was performed using a $30 \times 30$ mm$^2$ Peltier thermode (CHEPS Pathway, Medoc) at three different intensities: low heat (42°C), medium heat (46°C), and high heat (48°C) at the left radial forearm. These three temperatures cover a large range of temperatures associated with nociception. The lowest temperature was set at 42°C to ensure a temperature above the median threshold of heat-sensitive C-fiber nociceptors which have a median heat threshold of 41°C (*Treede et al., 1998*). The baseline temperature was set at 33°C and the rise rate to 40°C/s. After two blocks, the stimulated skin patch was changed to avoid sensitization.

Aversive pictures were chosen from the International Affective Picture System (IAPS) (*Lang et al., 2008*) database at three different levels of aversiveness. The images presented during the EEG experiment had three levels of valence of which the low valence category had valence values of $2.02 \pm 0.05$ (mean $\pm$ standard error), the medium valence category had valence values of $4.06 \pm 0.02$ (mean $\pm$ standard error), and the high valence category had valence values of $5.23 \pm 0.01$ (mean $\pm$ standard error).

Prior to each picture or heat stimulus, a visual cue was presented. The color of the cue (triangle) indicated (probabilistically) the modality of the stimulus (orange for picture and blue for heat). A white digit written inside of each triangle indicated (probabilistically) the intensity of the subsequent stimulus (1, 2, and 3 for low, medium, and high intensity, respectively). During the whole trial, a centered fixation cross was presented on the screen.

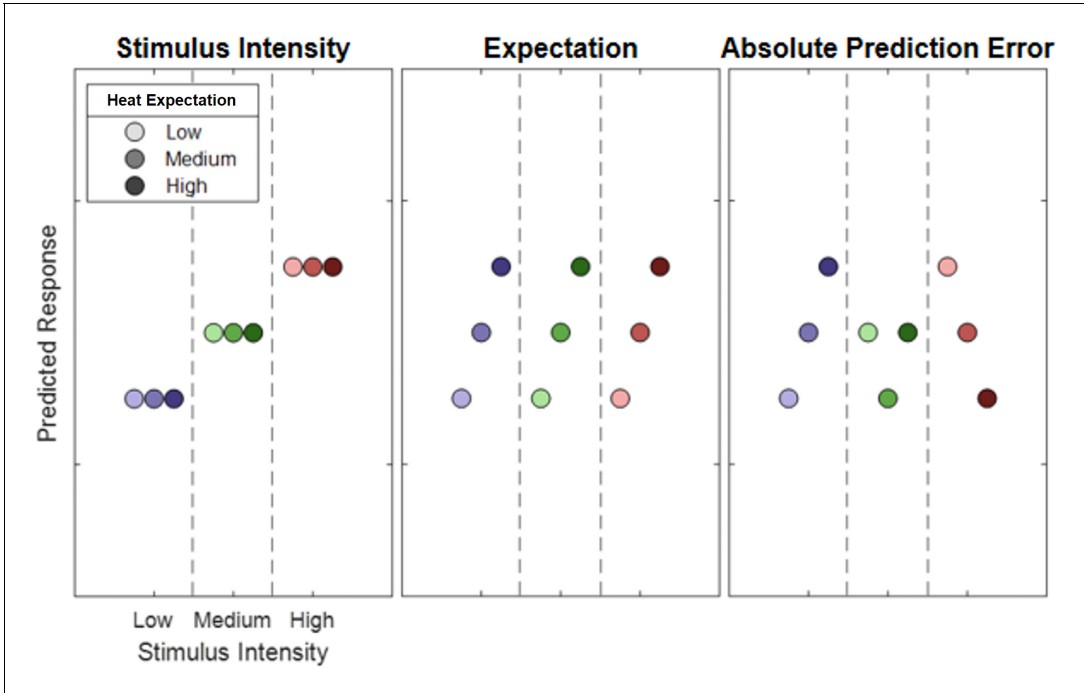

**Figure 2.** Hypothetical response patterns based on stimulus intensity (left), expectation (middle), and absolute prediction error (right). The y-axis represents a hypothetical response variable (e.g. electroencephalogram [EEG] power). Each dot represents a different condition for each stimulus–cue combination. Blue colors represent low heat conditions, green colors represent medium heat conditions, and red colors represent high heat conditions. Color intensities depict expectation level.

Each trial began with the presentation of the cue for 500 ms as an indicator for the modality and intensity of the subsequently presented stimulus. The modality was correctly cued in 70% of all trials by the color of the triangle. In 60% of all trials, the stimulus intensity was correctly indicated by the digit within the triangle (see *Figure 1* for an overview of all cue contingencies).

Before the presentation of the stimulus, there was a blank period with a variable time frame between 1000 and 1400 ms. Then, the visual or thermal stimulus was presented for a duration of 2 s. The visual stimulus was centered on the screen and allowed the participant to perceive the stimulus without eye movements. Right after the termination of the stimulus, subjects were asked to rate the aversiveness of the stimulus on a four-point rating scale, where one was labeled as 'neutral' and four was labeled as 'very strong'. Ratings were performed using a response box operated with the right hand (see *Figure 1* for a visualization of the trial structure).

In addition, four catch trials were included in each block. Subjects were asked to report the preceding cue in terms of their information content of the modality and intensity within 8 s, and no stimulation was given in these trials.

Trials were presented in four blocks. Each block consisted of 126 trials and four catch trials and lasted about 15 min. The trial order within each block was pseudorandomized. The order of blocks was randomized across subjects. The whole EEG experiment including preparation and instructions lasted for about 3 hr.

Prior to the actual EEG experiment, subjects participated in a behavioral training session. During this session, participants were informed about the procedure and gave their written informed consent. The behavioral training session was implemented to avoid learning effects associated with the contingencies between the cues and the stimuli during the EEG session. Between two and three blocks were presented during the training session (without electrophysiological recordings). The experimenter assessed the performance after each block based on the percentage of successful catch trials and the ability to distinguish the three levels of aversiveness of each modality. The training session was terminated after the second block if participants were able to successfully label cues in 75% of the catch trials within the second block.

## EEG data acquisition

EEG data were acquired using a 64-channel Ag/AgCl active electrode system (ActiCap64; BrainProducts) placed according to the extended 10–20 system (*Klem et al., 1999*). Sixty electrodes were used of the most central scalp positions. The EEG was sampled at 500 Hz, referenced at FCz, and grounded at Iz. For artifact removal, a horizontal, bipolar electrooculogram (EOG) was recorded using two of the remaining electrodes and placing them on the skin approximately 1 cm left from the left eye and right from the right eye at the height of the pupils. One vertical EOG was recorded using one of the remaining electrodes centrally approx. 1 cm beneath the left eyelid and another electrode was fixated on the neck at the upper part of the left trapezius muscle (Musculus trapezius) to record an electromyogram.

## EEG preprocessing

The data analysis was performed using the Fieldtrip toolbox for EEG/ MEG (magnetoencephalogram) analysis (*Oostenveld et al., 2011*) at Donders Institute for Brain, Cognition and Behaviour, Radboud University Nijmegen, the Netherlands (see http://www.ru.nl/neuroimaging/fieldtrip). EEG data were epoched and time-locked to the stimulus onset using the electrical trigger signaling the thermode to start the temperature rise of a given heat trial. Each epoch was centered (subtraction of the temporal mean) and detrended and included a time range of 3410 ms before and 2505 ms after trigger onset.

The data was band pass-filtered at 1–100 Hz, Butterworth, fourth order. EEG epochs were then visually inspected and trials contaminated by artifacts due to gross movements or technical artifacts were removed. Subsequently, trials contaminated by eye-blinks and movements were corrected using an independent component analysis (ICA) algorithm (*Makeig et al., 1996*; *Jung et al., 2000*). In all datasets, individual eye movements, showing a large EOG channel contribution and a frontal scalp distribution, were clearly seen in the removed independent components. Additionally, time–frequency decomposed ICA data were inspected at a single trial level, after z-transformation (only for artifact detection purposes) based on the mean and the standard deviation across all components separately for each frequency from 31 to 100 Hz. Time–frequency representations were calculated using a sliding window multi-taper analysis with a window of 200 ms length, which was shifted over the data with a step size of 20 ms with a spectral smoothing of 15 Hz. Artifact components or trials were easily visible and were compared with the raw ICA components. Specifically, single and separate muscle spikes were identified as columns or 'clouds' in time–frequency plots. Using this procedure, up to 30 components were removed before remaining non-artifactual components were back-projected and resulted in corrected data. Subsequently, the data was re-referenced to a common average of all EEG channels and the previous reference channel FCz was reused as a data channel (see *Figure 1—figure supplement 1* for a summary of rejected components per participant).

Before time–frequency transformations for data analysis were performed on the cleaned dataset, the time axis of single trials was shifted to create cue-locked and stimulus-locked data. Cue-locked data uses the onset of the cue as t = 0. Stimulus-locked data takes the ramp-up period of the thermode into account and sets t = 0 to the time point when the thermode reached the target temperature (225, 325, and 375 ms after trigger onset for low, medium, and high heat conditions, respectively). Frequencies up to 30 Hz (1–30 Hz in 1 Hz steps) were analyzed using a sliding Hanning-window Fourier transformation with a window length of 300 ms and a step size of 50 ms. For the analysis of frequencies higher than 30 Hz (31–100 Hz in 1 Hz steps), spectral analyses of the EEG data were performed using a sliding window multi-taper analysis. A window of 200 ms length was shifted over the data with a step size of 50 ms with a spectral smoothing of 15 Hz. Spectral estimates were averaged for each subject over trials. Afterward, a z-baseline normalization was performed based on a 500 ms baseline before cue onset. For cue-locked data, a time frame ranging from −650 to −150 ms was chosen as a baseline. A distance from the cue onset to the baseline period of 150 ms was set because of the half-taper window length of 150 ms, that is, data points between −150 and 0 ms are contaminated by the onset of the cue. For stimulus-locked trials, a variable cue duration (1500–1900 ms) and a variable stimulus offset based on the ramp-up time (225–375 ms) were additionally taken into account, resulting in an according baseline from −2950 to −2450 ms from stimulus onset. For the baseline correction of time–frequency data, the mean and standard deviation were estimated for the baseline period (for each subject–channel–frequency combination,

separately). The mean spectral estimate of the baseline was then subtracted from each data point, and the resulting baseline-centered values were divided by the baseline standard deviation (classical baseline normalization – additive model; see *Grandchamp and Delorme, 2011*).

## Predictive coding model

Similar to a previous fMRI study (*Fazeli and Büchel, 2018*), our full model included three experimental within-subject factors (see *Figure 2*). The stimulus intensity factor (INT) models the measured response with a simple linear function of the stimulus intensity (−1, 0, and 1 for low, medium, and high intensities, respectively). The expectation factor (EXP) was defined (see *Figure 2*; center column) linearly from the intensity predicted by the cue. Again, conditions with a low intensity cue were coded with a −1, conditions with a medium intensity cue with a 0 and conditions with a high intensity cue with a 1. The absolute prediction error factor (PE) resulted from the absolute difference of the expectation and actual stimulus intensity (see *Figure 2*; right column).

We also investigated a signed PE. However, it should be noted that such a term is not orthogonal to the EXP. However, assuming that an EXP can only be observed after the cue and a PE after the nociceptive stimulus, we were able to test for a signed PE during the heat phase. Also, we considered a one-sided PE, where a prediction error is only signaled when the stimulus is more intense as expected, which is motivated by previous work (*Egner et al., 2010*; *Summerfield and de Lange, 2014*; *Geuter et al., 2017*).

As the lowest stimulus intensity was often perceived as non-painful, we additionally performed an analysis only with medium and high stimulus intensities. Accordingly, the lowest stimulus intensity (42°C) were excluded in an additional repeated-measures ANOVA for this purpose (which will be referred to as the *reduced pain model*).

## Behavioral aversiveness ratings

Behavioral aversiveness ratings were averaged for all $3 \times 3$ cue–stimulus combinations over each participant, resulting in a $29 \times 9$ matrix (subject $\times$ condition) for the full model and a $29 \times 6$ matrix for the reduced pain model. We tested for main effects across stimulus intensity, expectation, as well as prediction error using a repeated-measures ANOVA as implemented in MATLAB (see fitrm and ranova; version 2020a, The MathWorks).

## EEG data analysis

All statistical tests in electrode space were corrected for multiple comparisons using non-parametrical permutation tests of clusters (*Maris and Oostenveld, 2007*). Cluster permutation tests take into account that biological processes are not strictly locked to a single frequency or time point and that activity could be picked up by multiple electrodes. Cluster permutation tests are specifically useful for explorative testing, as explained by *Maris and Oostenveld, 2007*. While prior hypotheses could have been formulated regarding the spatial, temporal, and spectral characteristics of brain responses associated with the intensity of thermal stimulation, and regions of interest could have been described, variations in the present design could be related to temporal and spectral differences compared to other studies, which would be taken into account using non-parametric cluster permutation testing.

We wanted to explore positive and negative time–frequency patterns associated with our variations of stimulus intensity, expectation, and absolute prediction errors using a repeated-measures ANOVA. A statistical value corresponding to a p-value of 0.05 ($F[1,28] = 4.196$) obtained from the repeated-measures ANOVA F-statistics of the respective main effect was used for clustering. Samples (exceeding the threshold of $F[1,28] = 4.196$) were clustered in connected sets on the basis of temporal (i.e. adjacent time points), spatial (i.e. neighboring electrodes), and spectral (i.e. $+/- 1$ Hz) adjacency. Further, clustering was restricted in a way that only samples were included in a cluster which had at least one significant neighbor in electrode space, that is, at least one neighboring channel also had to exceed the threshold for a sample to be included in the cluster. Neighbors were defined by a template provided by the Fieldtrip toolbox corresponding to the used EEG montage.

Subsequently, a cluster value was defined as the sum of all statistical values of included samples. Monte Carlo sampling was used to generate 1000 random permutations of the design matrix, and statistical tests were repeated in time–frequency space with the random design matrix. The

probability of a cluster from the original design matrix (p-value) was calculated by the proportion of random design matrices producing a cluster with a cluster value exceeding the original cluster. This test was applied two-sided for negative and positive clusters, which were differentiated by the average slope of the estimated factors.

Monte Carlo cluster tests were performed with 1000 permutations using the test statistics of a repeated-measures ANOVA model as the value for clustering (at p<0.05/F[1,28]=4.196). All tests were performed for low frequencies (1–30 Hz) and high frequencies (31–100 Hz), separately. Muscular and ocular electrodes were excluded from the cluster analysis.

The within-subject INT (which was coded as increasing linearly with stimulus intensity) was tested stimulus-locked from 0 to 1.6 s. The within-subject EXP, which was coded as increasing linearly with the cued stimulus intensity, was tested cue-locked from 0 to 3.6 s. The signed PE was coded as the difference between stimulus intensity and expectation. The absolute prediction error was coded as the absolute difference between stimulus intensity and expectation (see *Figure 2* for details). Additionally, we tested a one-sided prediction error, occurring only when the actual stimulus is of a higher intensity than expected. The signed, absolute, and one-sided PEs were tested stimulus-locked from 0 to 1.6 s.

## Results

### Behavioral data – aversiveness ratings

Participants experienced aversive heat or saw picture stimuli which were probabilistically cued in terms of modality and intensity, evoking an expectation of modality and intensity. The subsequently applied stimuli were then rated on a visual analog scale (VAS) from 1 to 4. Our primary behavioral question was whether ratings are influenced by the experimental manipulation of stimulus intensity, expectation, and absolute prediction errors.

To validate our intensity manipulation for thermal stimuli and to verify the discriminability between different levels of heat, we first tested for the main effect of stimulus intensity (*Figure 3a*). Our data show a clear rating difference between the three levels of heat. Results regarding the aversive pictures are not the focus of this report but are depicted in *Figure 3b* for the sake of comparison.

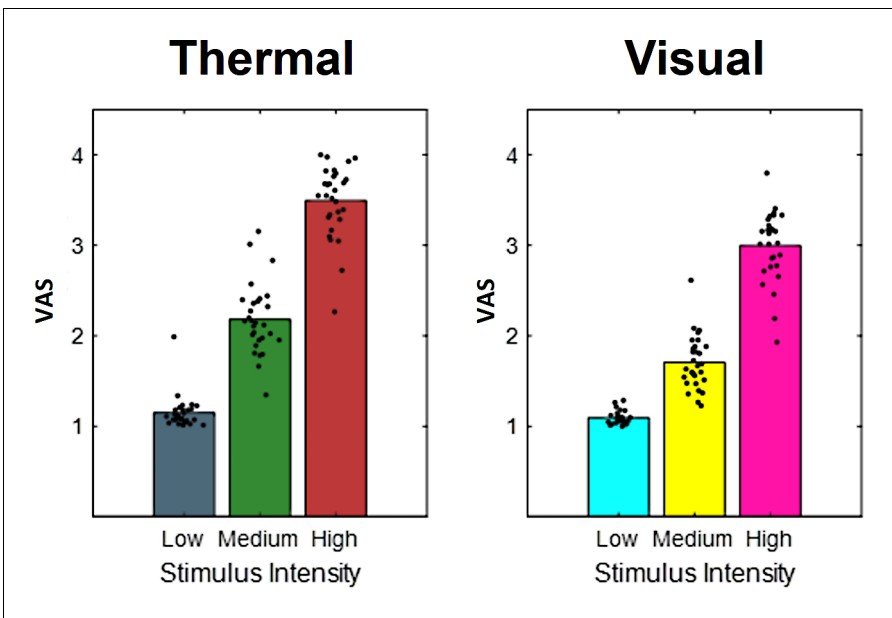

**Figure 3.** Bars indicate pooled aversiveness ratings for (**a**) heat and (**b**) aversive pictures for low-, medium-, and high-intensity conditions. Dots indicate average single-subject ratings.

To evaluate the main effects of stimulus intensity, expectation, and absolute prediction errors, we employed a repeated-measures ANOVA of the behavioral data, which revealed significant effects for the main effect of stimulus intensity, that is, the three levels of heat (F[1,28] = 743.97, p<0.001). Also, the main effect for expectation on aversiveness ratings was significant (F[1,28] = 38.53, p<0.001) (*Table 1*), indicating an influence of the cued intensity on behavioral aversiveness ratings (*Figure 4*). The absolute difference between the cued intensity and the actual stimulus intensity (i.e. absolute prediction error) only showed a trend effect on aversiveness ratings (F[1,28] = 2.87, p=0.10). The results regarding the aversive pictures are summarized in *Table 1*.

## EEG – stimulus intensity

In a first EEG analysis, we tested for a main effect of the intensity of the heat input in the context of a correctly cued modality (i.e. heat was expected and received). In order to do so, we performed a repeated-measures ANOVA on the time–frequency representation of the EEG data on low frequencies (1–30 Hz) and high frequencies (31–100 Hz) separately after stimulus onset using a cluster correction criterion to address the multiple comparisons problem (see 'Materials and methods' for details). Any significant cluster – composed of neighboring data points in time, frequency, and space – would indicate a neuronal oscillatory representation of variations in stimulus intensity in a given frequency band.

In the low frequency range (1–30 Hz), one positive cluster (i.e. a positive average slope of the factor) and one negative cluster (i.e. a negative average slope of the factor) were significant (*Figure 5*), indicating a linear association of stimulus intensity and power in this frequency range. Specifically, the negative cluster included samples in a time range from 250 to 1600 ms after stimulus onset in a frequency range from 1 to 30 Hz, predominately at contralateral central electrode sites (p=0.002). The highest parametric F-value within this cluster obtained from the repeated-measures ANOVA was F(1,28) = 36.40 (p<0.001). This sample was observed at 1250 ms and 22 Hz and had a maximum at channel CP2. All channels included samples of the negative low frequency stimulus intensity cluster.

Also in the low frequency range (1–30 Hz), a positive significant cluster included samples in a time range from 150 to 1050 ms after stimulus onset in the theta frequency range from 1 to 7 Hz predominately at midline electrode sites (p=0.048). The highest parametric F-value from the repeated-measures ANOVA was F(1,28) = 27.93 (p<0.001). This sample was found at 550 ms and 3 Hz and had a maximum at channel O2. All channels except FC5, CP4, C6, and FT7 were part of the positive low frequency stimulus intensity cluster.

In the high frequency range (31–100 Hz) representing gamma activity, one positive cluster was significant (p<0.001). This cluster included samples in a time range from 550 to 1600 ms after stimulus onset and frequencies from 46 to 100 Hz, predominately at contralateral centroparietal electrode sites (*Figure 5*). The highest parametric F-value within this cluster obtained from the repeated-measures ANOVA was F(1,28) = 33.35 (p<0.001). This sample was observed at 1600 ms and 100 Hz and had a maximum at channel Cz. All channels included samples of the positive high frequency stimulus intensity cluster.

In conclusion, these results indicate that a higher intensity of the thermal input is associated with increased theta and gamma band power and a negative relationship of alpha-to-beta band power and the intensity of the thermal input (see *Figure 5* for a summary of the results of the main effect

**Table 1.** Main effects of stimulus intensity, expectation, and absolute prediction errors on subjective ratings in both heat and picture conditions.

| Factor | Stimulus intensity (INT) | | Cued intensity (EXP) | | Absolute prediction error (PE) | |
|---|---|---|---|---|---|---|
| | F(1,28) | p | F(1,28) | p | F(1,28) | p |
| Modality | | | | | | |
| Thermal | 743.97 | <0.001 | 39.53 | <0.001 | 2.87 | 0.10 |
| Visual | 762.10 | <0.001 | 1.46 | 0.24 | 7.7 | 0.01 |

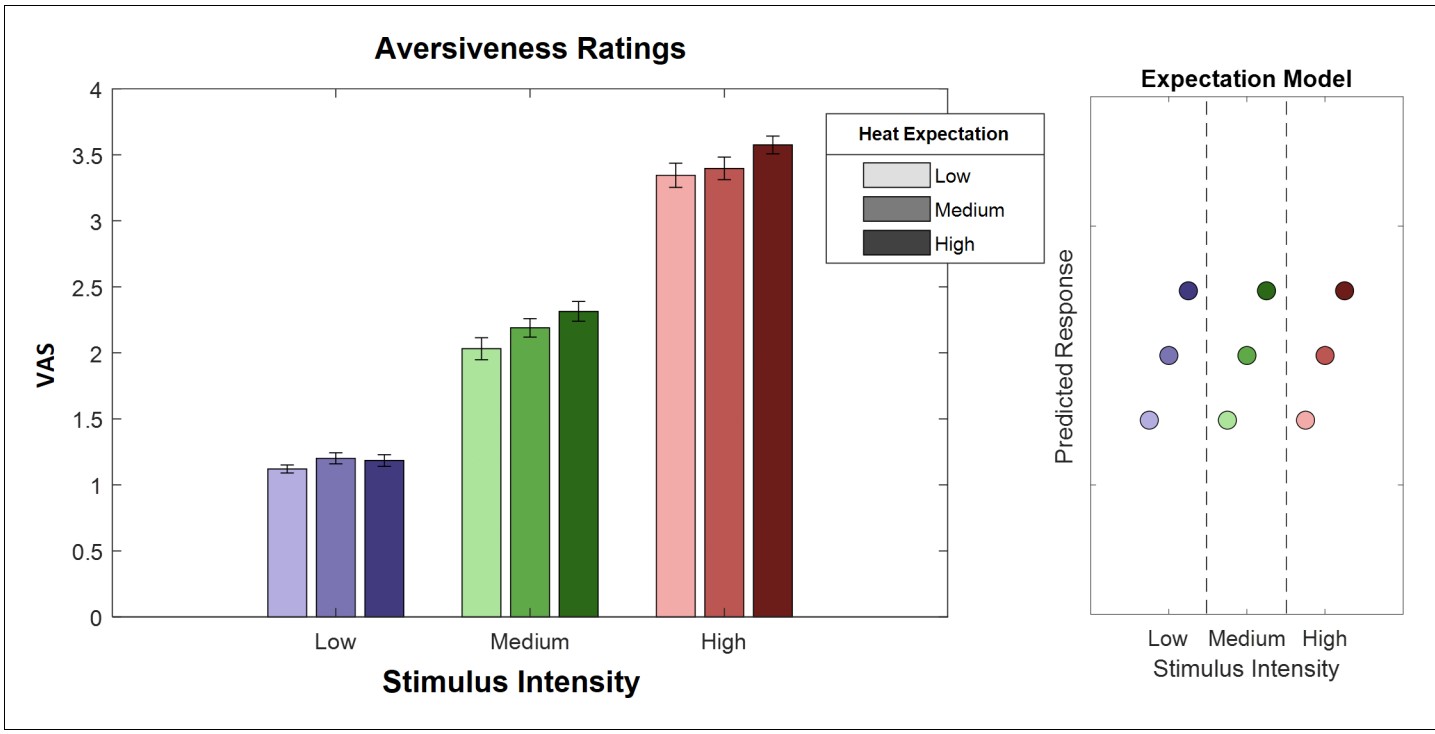

**Figure 4.** Ratings for heat stimuli (left) and 'expectation factor' weights (right). Bars indicate average aversiveness ratings. Ratings were given on a scale from 1 to 4. Error bars depict SEM. The data shows not only an effect of stimulus intensity (increase from blue to green to red) but also an effect of expectation (low to medium to high expectation). The right figure represents hypothetical response patterns based on the expectation factor. The y-axis represents the hypothetical response variable (e.g. visual analog scale [VAS] rating). Each dot represents a different condition for each stimulus–cue combination. Blue colors represent low heat conditions, green colors represent medium heat conditions, and red colors represent high heat conditions. Color intensities depict expectation level.

of stimulus intensity; see *Figure 5—figure supplement 1* for single-subject differences in the gamma band between low stimulus intensity and high stimulus intensity trials).

## Expectation

In a next step, we investigated the representation of EXP in our repeated-measures model, again for low frequencies (1–30 Hz) and high frequencies (31–100 Hz) separately in the cue-locked time–frequency representation of the EEG data.

This analysis revealed one significant positive cluster in the low frequency range (1–30 Hz), indicating a linear association of cue intensity (EXP) and power in this frequency range (p<0.05). The expectation cluster (p=0.022) included samples from time points ranging from 100 to 2000 ms after cue onset and included frequencies from 1 to 20 Hz. The highest parametric statistical test value (F [1,28] = 26.96, p<0.001) was observed at channel P1 700 ms after cue onset at a frequency of 9 Hz. All channels except TP8 included samples of the late expectation cluster (see *Figure 6* for a summary of the results of the expectation cluster; see *Figure 6—figure supplement 1* for single-subject values).

In summary, these results suggest an increase in alpha-to-beta band power to be associated with our experimental manipulation of expectations regarding the intensity of the thermal input.

## Prediction error model

Likewise, clustering was performed for the prediction error term after stimulus onset in low (1–30 Hz) and high frequencies (31–100 Hz). Any significant cluster would associate oscillatory activity with the difference of the expectation regarding the intensity of the thermal stimulation and the actual stimulation, representing a violation of this expectation (prediction error).

This analysis revealed a significant negative cluster in the high frequency range (31–100 Hz), indicating a (negative) linear association of absolute prediction errors and power in this frequency range

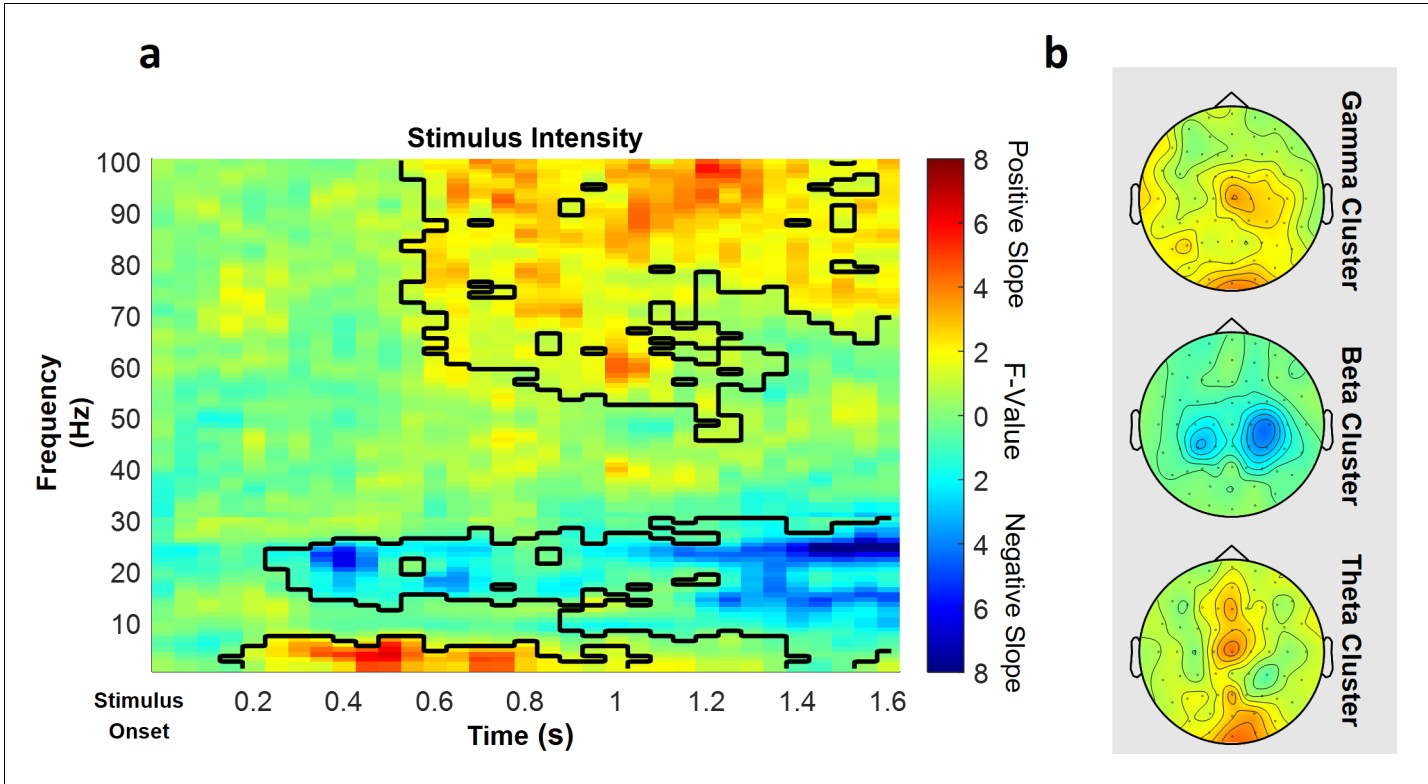

**Figure 5.** Parametric effects of stimulus intensity. Time–frequency representation averaged over all channels including a significant time–frequency sample of any cluster (**a**) and topographies over the whole cluster extents (i.e. full time and frequency range), respectively (**b**), of the stimulus intensity main effect of the repeated-measures ANOVA depicting increases (warm) and decreases (cold) in power in relation to heat stimulus intensity. Significant clusters are highlighted. Colors represent F-values from the repeated-measures ANOVA statistics for the main effect of stimulus intensity.
The online version of this article includes the following figure supplement(s) for figure 5:

**Figure supplement 1.** Difference for the main effect of stimulus intensity in the gamma band (averaged over 60–100 Hz, 1250–1600 ms) in power values for all high heat vs. low heat conditions with a valid modality cue (expect heat receive heat) for each subject, respectively.

(p=0.002). This (negative) absolute prediction error cluster included samples from frequencies ranging from 51 to 100 Hz and time points ranging from 50 to 1600 ms after stimulus onset. The highest parametric statistical test value (F(1,28) = 28.52, p<0.001) was found at channel O1 1300 ms after stimulus onset at a frequency of 98 Hz. All channels included samples of the absolute prediction error cluster (see *Figure 7* for a summary of the results; see *Figure 7—figure supplement 1* for single-subject values).

A cluster analysis of the signed prediction error, stimulus-locked after stimulus onset (from 1 to 30 Hz for low frequencies and 31–100 Hz for gamma frequencies; from 0 to 1600 ms, stimulus-locked), did not reveal any significant cluster of activity associated with a linear increase or decrease of EXP (all p>0.05). Ignoring all stimulus–cue combinations of the PE where the stimulus intensity was less intense than expected leads to a one-sided PE. A cluster analysis of this effect did not reveal any significant cluster of activity (all p>0.05).

In summary, these results suggest a decrease in gamma band power to be associated with our experimental manipulation of expectation violations, resulting from a mismatch of the cued intensity and the actual heat input.

## Reduced pain model

In an additional analysis, we tested all effects in a reduced pain model, which only included painful stimuli (i.e. three expectation levels and two intensity levels).

To evaluate the main effects of stimulus intensity, expectation, and absolute prediction errors in the behavioral data, we employed again a repeated-measures ANOVA which revealed significant

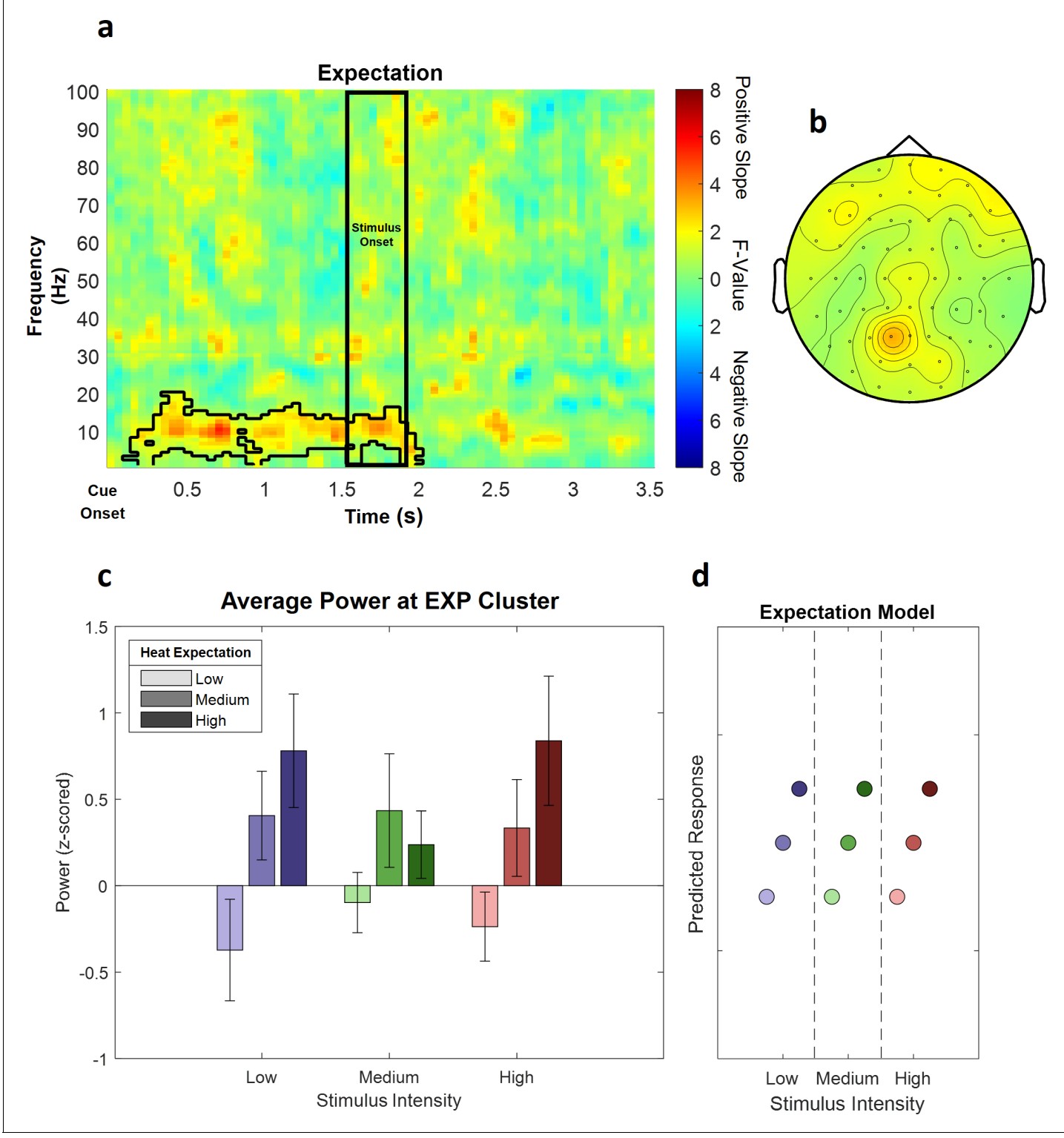

**Figure 6.** The main effect of expectation. (a) Time–frequency representation of the statistical F-values averaged over all channels. The significant cluster is highlighted. The black box between 1500 and 1900 ms marks the jittered onset of the trigger signal to start the ramp-up of the heat stimulus. (b) Topography of the averaged power over time and frequency of the whole cluster extent (i.e. over the whole time and frequency range) at each channel. Brighter colors indicate higher F-values. (c) Power values for all conditions with a valid modality cue (expect heat receive heat) averaged over all significant time–frequency–electrode samples of the EXP cluster show alpha-to-beta enhancement (i.e. positive representation) associated with expectation. Error bars represent SEM. (d) Predicted responses based on the positive expectation factor are shown. The y-axis represents an imaginary

*Figure 6 continued on next page*

*Figure 6 continued*

response variable (e.g. EEG power). Each dot represents a different condition (in the order of the bar plot representation of average EEG power) for each stimulus–cue combination. Blue colors represent low heat conditions, green colors represent medium heat conditions, and red colors represent high heat conditions. Color intensities depict expectation level.

The online version of this article includes the following figure supplement(s) for figure 6:

**Figure supplement 1.** Power values for all conditions with a valid modality cue (expect heat receive heat) averaged over all significant time–frequency–electrode samples period for each subject (ID) of the EXP cluster.

effects for the main effect of stimulus intensity, that is, the two remaining levels of pain ($F_{[1,28]}$ = 1109.9, p<0.001). Also, the main effect for expectation on pain ratings was significant ($F_{[1,28]}$ = 17.07, p<0.001), indicating again an influence of the cued intensity on behavioral pain ratings. The absolute difference between the cued intensity and the actual stimulus intensity (i.e. absolute prediction error) when only painful stimuli were included revealed a positive significant effect on pain ratings ($F_{[1,28]}$ = 80.75, p<0.001). This indicates prediction errors and prior expectations to modulate behavioral aversiveness ratings in painful stimulation.

For the analysis of time–frequency EEG data, we performed a repeated-measures ANOVA on the time–frequency representation of the EEG data on low frequencies (1–30 Hz) and high frequencies (31–100 Hz) separately and again using the same cluster correction criterion to address the multiple comparisons problem as in the initial analysis of the full model.

The cluster test of stimulus intensity revealed one negative cluster (p=0.014) in the low frequency range (1–30 Hz) including time points from 850 to 1600 ms and frequencies from 8 to 30 Hz (*Figure 8a*; see *Figure 8—figure supplement 1* for single-subject values). The maximum statistical F-value ($F_{[1,28]}$ = 31.82; p<0.001) was found at channel AF3 at a frequency of 30 Hz at 1600 ms and revealed a similar but more broad topography as compared to the original alpha-to-beta negative main effect of stimulus intensity of the analysis of the full model. All channels included samples of the negative stimulus intensity cluster.

In the high frequency range (31–100 Hz), a negative cluster of activity (p=0.038) was associated with absolute prediction errors and included samples in a time range from 850 to 1600 ms after stimulus onset in the gamma frequency range from 54 to 90 Hz predominately at occipital and parietal electrode sites. The highest parametric F-value from the repeated-measures ANOVA was $F_{(1,28)}$ = 24.10 (p<0.001). This sample was found at 1150 ms and 77 Hz and had a maximum at channel F8 (*Figure 8b*; see *Figure 8—figure supplement 2* for single-subject values). All channels except FC5, CP4, C6, and FT7 were part of the gamma frequency negative absolute prediction error cluster.

In the low (1–30 Hz) and high frequency (31–100 Hz) ranges, no significant cluster was observed representing a significant relationship between expectations and EEG activity. However, one cluster in the low frequency range (1–30 Hz) showed a trend level (p=0.14; based on cluster mass, i.e., the sum of all clustered F-values) and included samples in a time range from 550 to 1600 ms after stimulus onset and frequencies from 6 to 24 Hz and is displayed in dotted lines in *Figure 8c* (see *Figure 8—figure supplement 3* for single-subject values).

## Discussion

Using a cued heat paradigm with three different stimulus intensities, our data showed a clear discriminability of different levels of aversiveness based on behavioral ratings and EEG time–frequency patterns. Specifically, we observed several clusters of activity to be associated with the intensity of thermal stimulation in the theta, beta, and gamma band. Furthermore, behavioral data clearly indicated a positive influence of cued intensity on pain perception. In addition, our results provide evidence for temporally and spectrally separable clusters of oscillatory activity associated with expectation and a negative modulation of gamma activity by prediction errors for thermoception and pain. Specifically, one early low frequency (1–30 Hz) cluster was related to expectation in thermoception, that is, cued intensity. In contrast, a later occurring cluster at higher frequencies (31–100 Hz) was related to negative prediction errors in thermoception and pain.

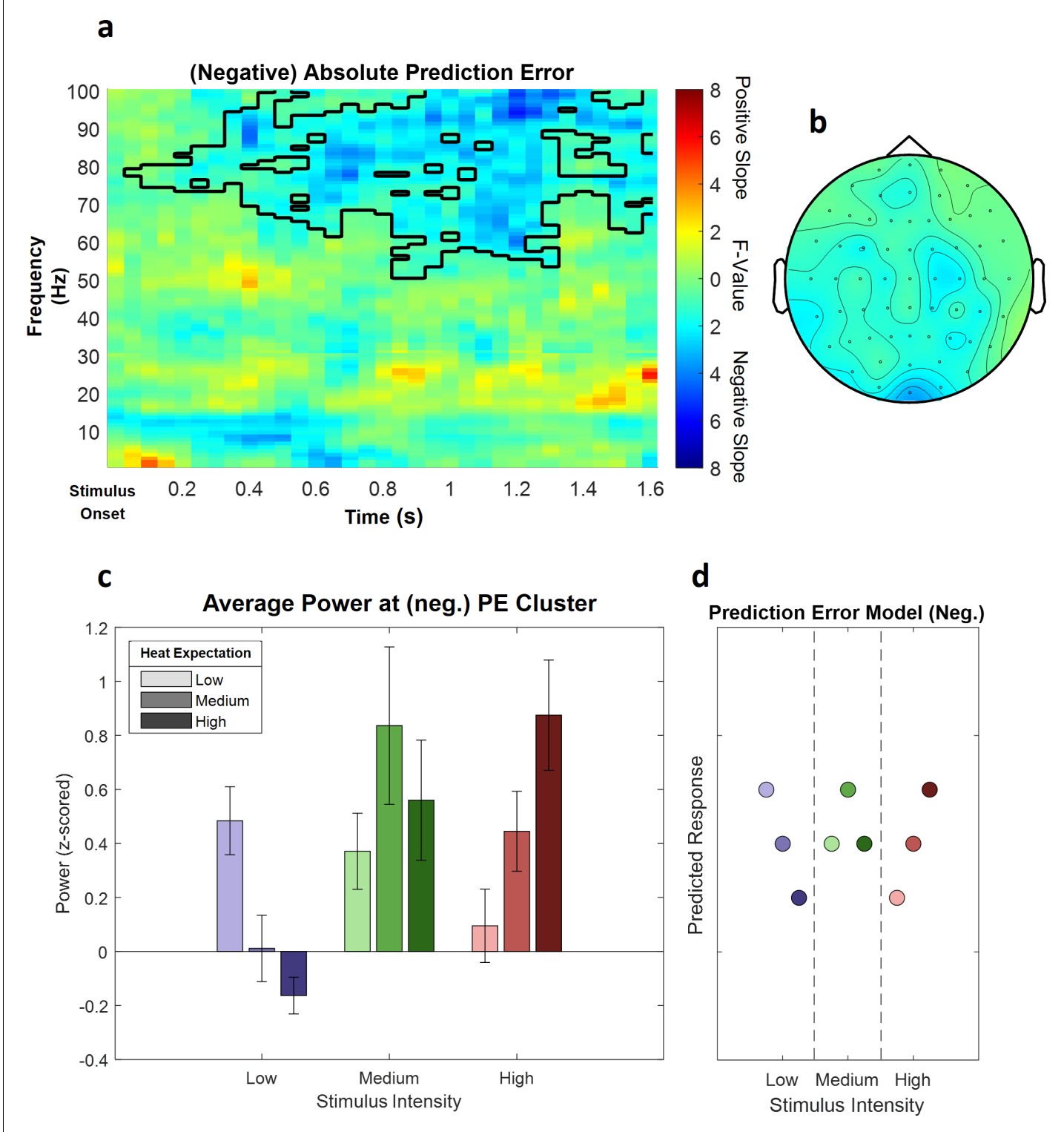

**Figure 7.** The main effect of absolute prediction errors. (**a**) Time–frequency representation of the statistical F-values averaged over all channels. The significant cluster is highlighted. (**b**) Topography of the averaged power over time and frequency of the whole cluster extent (i.e. over the whole time and frequency range) at each channel. Brighter colors indicate higher F-values. (**c**) Power values for all conditions with a valid modality cue (expect heat receive heat) averaged over all significant time–frequency–electrode samples of the prediction error factor (PE) cluster show gamma decreases (i.e. negative representation) associated with prediction errors. Error bars represent SEM. (**d**) Predicted responses based on the negative PE are shown: The y-axis represents an imaginary response variable (e.g. electroencephalogram [EEG] power). Each dot represents a different condition (in the order of

*Figure 7 continued on next page*

*Figure 7 continued*

the bar plot representation of average EEG power) for each stimulus–cue combination. Blue colors represent low heat conditions, green colors represent medium heat conditions, and red colors represent high heat conditions. Color intensities depict expectation level.

The online version of this article includes the following figure supplement(s) for figure 7:

**Figure supplement 1.** Power values for all conditions with a valid modality cue (expect heat receive heat) averaged over all significant time–frequency–electrode samples period for each subject (ID) of the negative absolute prediction error cluster.

## Stimulus intensity and oscillatory activity

Note that our definition of stimulus onset is based on the moment the thermode reached the target temperature. Using a thermode heating gradient of 40°C/s and neglecting any small internal delays, the target temperatures of 42°C, 46°C, and 48°C are reached after 225, 325, and 375 ms, respectively. Therefore, our observed increase in theta power agrees with previous studies (*Ploner et al., 2017*) and most likely correspond to pain-related evoked potentials (*Lorenz and Garcia-Larrea, 2003*; *Tiemann et al., 2015*), such as the P2 with a similar topography. In addition, we observed a significant suppression of alpha-to-beta activity which, given the abovementioned delays of our painful stimuli, is in line with the reported beta suppression in previous EEG studies on pain (*Mouraux et al., 2003*; *Ploner et al., 2006*; *May et al., 2012*; *Hu et al., 2013*). Finally, power in the gamma band was also correlated with heat intensity, which is in line with previous studies (*Gross et al., 2007*; *Hauck et al., 2007*; *Zhang et al., 2012*; *Rossiter et al., 2013*; *Tiemann et al., 2015*). Interestingly, only the alpha-to-beta band desynchronization differentiated between medium and high pain conditions, whereas differences in the theta and gamma band activity were only evident when the lowest stimulus intensity was included which was perceived as neutral.

We observed a behavioral effect of prediction errors on perceived stimulus intensity in the reduced pain model, but this effect was only a trend in the full model. The latter finding replicates a previous study (*Fazeli and Büchel, 2018*) indicating a robust effect. Interestingly, the effect of prediction errors on perception increased, and became significant, when we constrained our analysis to the clearly painful stimuli (reduced pain model). This suggests that a prediction error seems to more strongly affect pain perception, whereas the effect is weaker in the context of thermoception. However, this speculation should be corroborated in a future study.

On a more conceptual level, the investigation of neurophysiological effects even in the absence of a behavioral effect has been considered meaningful (*Wilkinson and Halligan, 2004*). In particular, the authors argue that because it is commonly unknown which parts of a cognitive process (and in which way) produce a specific behavioral response, the relationship between neurophysiological data and behavioral responses should not be overemphasized, and therefore it can be misleading to declare behavioral effects a reference or 'gold standard'. Studies aiming to understand neurophysiological mechanisms of cognition usually relate a neurophysiological readout to a known perturbation (i.e. experimental design), which is meaningful in its own right.

## Hypotheses based on microcircuits

Theoretical accounts (*Arnal and Giraud, 2012*; *Bastos et al., 2012*) have suggested that predictive coding mechanisms could be related to the functional architecture of neuronal microcircuits. As feedforward connections are predominately originating from superficial layers and feedback connections from deep layers, it has been suggested that prediction errors should be expressed by higher frequencies than the predictions that accumulate them.

In the auditory modality, these ideas are supported by empirical data (*Todorovic et al., 2011*) showing that prediction errors in the context of repetition suppression were associated with higher gamma band activity. Likewise, in the visual domain, an MEG study has shown that temporo-parietal beta power was correlated with the predictability of an action kinematics–outcome sequence, while gamma power was correlated with the prediction error (*van Pelt et al., 2016*).

## Frequency patterns in predictive coding of pain

Only a few studies have investigated the spectral and temporal properties of expectations and prediction errors in the context of pain (summarized by *Ploner et al., 2017*). A recent study in rodents has suggested an information flow between S1 gamma and ACC (Anterior Cingulate Cortex) beta

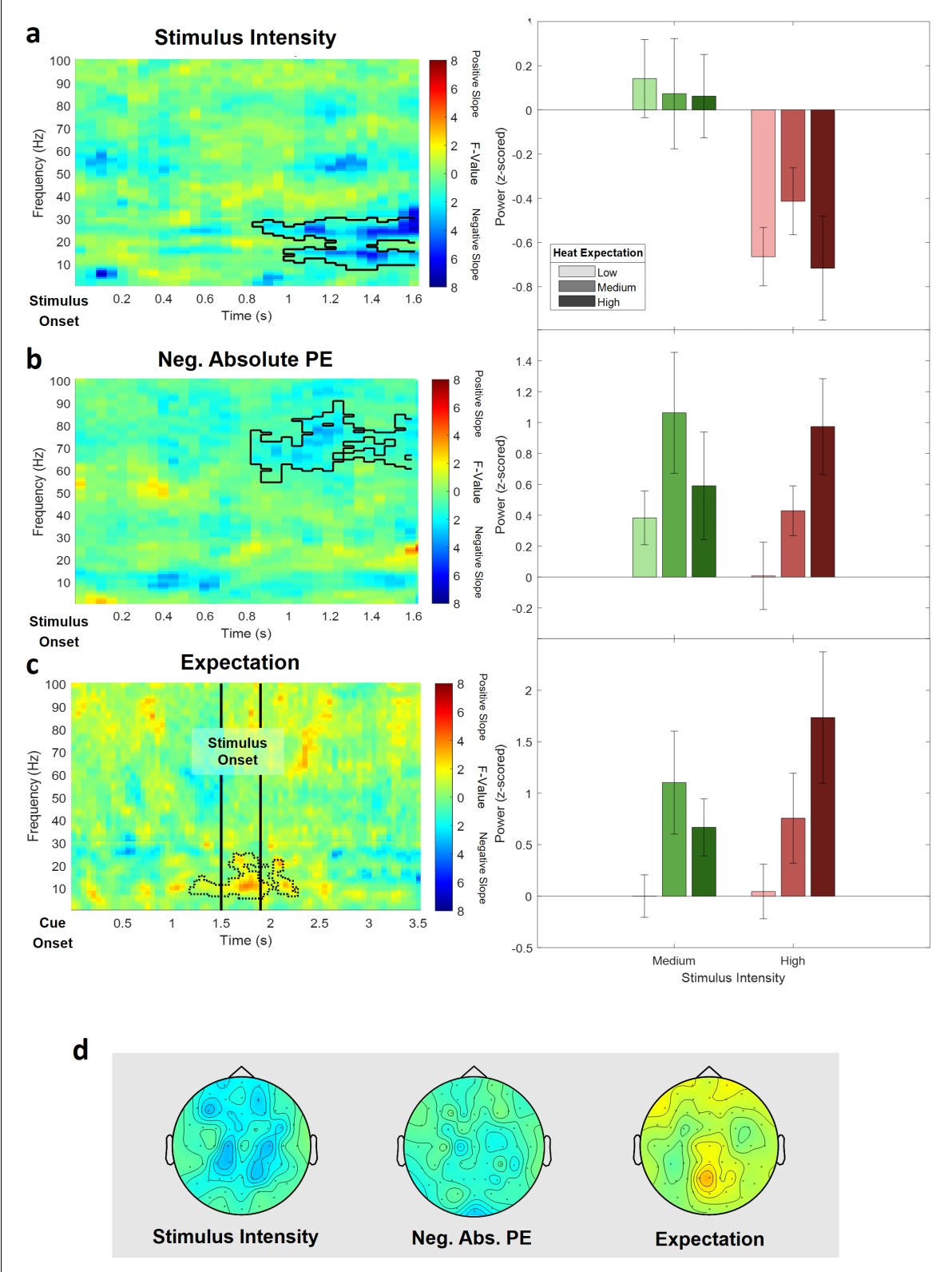

**Figure 8.** Electroencephalogram (EEG) data analysis of the reduced pain model. The top three rows show (a) the main effect of stimulus intensity, (b) the main effect of negative absolute prediction errors, and (c) the main effect of expectation. Left column: time–frequency representation of the statistical F-values averaged over all channels. Significant clusters are highlighted by a solid line. The non-significant expectation cluster is highlighted by a thin dotted line. Right column: power values for all conditions included in the reduced model with a valid modality cue (expect heat receive heat)

*Figure 8 continued on next page*

*Figure 8 continued*

averaged over all significant time–frequency–electrode samples of the respective cluster. (**d**) Topographies of the averaged power over time and frequency of the whole cluster extent (i.e. over the whole time and frequency range) at each channel for stimulus intensity (left), negative absolute prediction errors (center), and expectation (right). Brighter colors indicate higher F-values.

The online version of this article includes the following figure supplement(s) for figure 8:

**Figure supplement 1.** Power values for all medium and high intensity conditions with a valid modality cue.
**Figure supplement 2.** Power values for all medium and high intensity conditions with a valid modality cue.
**Figure supplement 3.** Power values for all medium and high intensity conditions with a valid modality cue.

activity during spontaneous pain (*Xiao et al., 2019*). Based on these data, the authors have proposed a predictive coding model including a bottom-up (gamma) and top-down (beta) component (*Song et al., 2019*). Finally, in humans, a recent EEG study showed that the sensorimotor cortex is more strongly connected to the medial prefrontal cortex at alpha frequencies during tonic pain, suggesting alpha band activity in tonic pain to be associated with bottom-up instead of top-down signaling (*Nickel et al., 2020*). Nevertheless, the focus of these studies was on generic interactions (i.e. top-down vs. bottom-up) processes without directly inducing prediction errors as in a cued pain paradigm employed in our study.

In the flexible routing model proposed by *Ploner et al., 2017*, pain is seen as driven by contextual processes, such as expectations, which is associated with alpha/beta oscillations and alpha/beta synchrony across brain areas. Previous studies have started to examine the spectral properties of mechanisms related to generative models of pain perception. In particular, a previous MEG study reported that alpha suppression in the anterior insula is related mainly to pain expectation in a paradigm in which painful stimuli were interleaved with non-painful stimuli (*Franciotti et al., 2009*). This was interpreted as a preparatory mechanism for an upcoming painful stimulus. In a related study, alpha desynchronization in the context of predictable painful stimuli has been discussed as a possible neural correlate of attentional preparatory processes (*Babiloni et al., 2003*).

Expectation is also a crucial ingredient of placebo analgesia and nocebo hyperalgesia. A previous study reported that resting-state alpha band activity was also linked to the expectation of pain modulation (analgesia) in a placebo paradigm (*Huneke et al., 2013*). With respect to negative expectations, it has been shown that pain modulation due to nocebo expectation is associated with enhanced alpha activity (*Albu and Meagher, 2016*). Our findings are in line with these results indicating an important role of low frequency activity in mediating expectation effects in a pain network underlying a generative model for pain perception.

In contrast to prediction error effects in the visual (*Bauer et al., 2014*; *van Pelt et al., 2016*) and auditory (*Edwards et al., 2005*; *Parras et al., 2017*) domains, we observed a negative modulation of gamma activity by absolute prediction errors. However, it should be noted that opposite effects have been observed in other cognitive domains. For instance, increased gamma power has been associated with successful matching (i.e. the absence of a prediction error) between external input and internal representation (*Herrmann et al., 2004a*; *Osipova et al., 2006*; *Wang et al., 2018*). In particular, gamma band responses have been explained in terms of the match between bottom-up and top-down information (*Herrmann et al., 2004b*). One example is the observation of increased gamma activity with a higher so-called *cloze probability* in sentence-level language comprehension (*Hald et al., 2006*; *Obleser and Kotz, 2011*; *Wang et al., 2012*; *Wang et al., 2018*; *Molinaro et al., 2013*). It has been shown that a critical word that is semantically predictable by the preceding sentence (so-called high cloze probability) induces a larger gamma response than words which are semantically incongruent (i.e. unpredicted; low cloze probability) (*Wang et al., 2018*).

## Pain vs. thermoception

In the present study, the lowest stimulus intensity was often not perceived as painful but as hot. In general, stimulus properties were chosen to be comparable to a previous fMRI study which showed fMRI signals related to prediction errors (*Fazeli and Büchel, 2018*). However, even though the lowest stimulus intensity (42°C) was above the threshold of nociceptors (*Treede et al., 1998*), the subjective experience of the lowest pain stimuli was often rated as neutral. Therefore, we performed an additional analysis (reduced pain model) only comprising clearly painful stimuli (46°C and 48°C) to

more specifically address expectations and predictions errors in pain. The analyses of the behavioral data revealed similar results. Both models showed a highly significant effect of stimulus intensity and expectation on perceived stimulus intensity. In addition, the reduced pain model showed a significant prediction error effect, which was formally not observed in the full model. However, it is important to note that this difference should not be overinterpreted, as the p-value for the prediction error effect of the full model was at a trend level (p=0.1). Importantly, the negative representation of prediction errors in the gamma band was evident in both, the reduced and the full model.

## Limitations

To unravel the temporal aspects of expectations and prediction errors, this study has been designed in close analogy to a previous fMRI study and we decided to use the same experimental paradigm (*Fazeli and Büchel, 2018*). We therefore decided to also keep the sample characteristics similar and restricted the sample to male participants, which means that we cannot generalize our results to the population. However, our study agrees with the findings of a previous study using a similar design (*Geuter et al., 2017*) which tested male and female participants. Future studies should investigate samples including female participants. This would also allow to investigate sex effects with respect to expectation and prediction error effects in pain.

To minimize motor responses and speed up the rating procedure, we used a four-button device to directly assess stimulus intensity (in contrast to using two buttons to move a slider on a VAS), thus being limited to a coarse rating scale of four levels, where one was labeled as 'neutral' and four was labeled as 'very strong'. This allows to accommodate more trials but is not ideal to assess fine-grained differences, specifically to differentiate between non-painful and painful stimulation, as level 1 would represent 0–25 on a 0–100 VAS. Future research could use conventional 0–100 VAS to assess stimulus intensity on a finer scale.

For reasons of comparability to a previous fMRI study, we employed three different temperatures for all volunteers. Alternatively, we could have defined three levels of pain based on individual calibration of heat stimuli (*Taesler and Rose, 2017*; *Grahl et al., 2018*; *Horing et al., 2019*; *Zhang et al., 2020*; *Feldhaus et al., 2021*). Such a procedure could have avoided trials where no pain was subjectively perceived. On the other hand, such an approach also carries the risk that subjective ratings during the calibration process do not truly reflect pain and can lead to errors (especially if ratings are too low) which then affect the entire experiment. However, to address this shortcoming, we performed an additional analysis, which only included painful stimulus intensities.

## Summary

Our data show that key variables required for pain perception and thermoception in the context of a generative model are correlated with distinct oscillatory profiles in the brain. Furthermore, each oscillatory frequency band was correlated with a distinct variable such as expectation and prediction errors. These mechanistic insights could be very helpful in patients with acute and more importantly in patients with chronic pain, where expectations have been shown to play a critical role in pain persistence.

## Acknowledgements

We would like to thank Markus Ploner for comments on an earlier version of this manuscript. We would also like to thank Matthias Kerkemeyer for his help during data collection. CB is supported by DFG SFB 289 project A02 and ERC-AdG-883892-PainPersist. MR is supported by DFG SFB 289 project A03 and DFG SFB TR 169 project B3. Funded by the Deutsche Forschungsgemeinschaft (DFG, German Research Foundation) – Project-ID 422744262–TRR 289.

## Additional information

### Funding

| Funder | Grant reference number | Author |
| --- | --- | --- |
| Deutsche Forschungsgemeinschaft | 422744262–TRR 289 project A02 | Christian Büchel |

| Deutsche Forschungsge-meinschaft | DFG SFB TR 169 project B3 | Michael Rose |
|---|---|---|
| H2020 European Research Council | ERC-AdG-883892-PainPersist | Christian Büchel |
| Deutsche Forschungsge-meinschaft | 422744262–TRR 289 project A03 | Michael Rose |

The funders had no role in study design, data collection and interpretation, or the decision to submit the work for publication.

### Author contributions

Andreas Strube, Conceptualization, Data curation, Software, Formal analysis, Investigation, Visualization, Methodology, Writing - original draft, Project administration, Writing - review and editing; Michael Rose, Conceptualization, Resources, Software, Methodology, Writing - review and editing; Sepideh Fazeli, Conceptualization, Software, Supervision; Christian Büchel, Conceptualization, Resources, Formal analysis, Supervision, Funding acquisition, Validation, Visualization, Methodology, Project administration, Writing - review and editing

### Author ORCIDs

Andreas Strube (iD) https://orcid.org/0000-0002-6545-0366
Christian Büchel (iD) https://orcid.org/0000-0003-1965-906X

### Ethics

Human subjects: All volunteers gave their informed consent. The study was approved by the Ethics board of the Hamburg Medical Association (PV4745).

### Decision letter and Author response

Decision letter https://doi.org/10.7554/eLife.62809.sa1
Author response https://doi.org/10.7554/eLife.62809.sa2

## Additional files

### Supplementary files

• Transparent reporting form

### Data availability

Data for this study are available on https://osf.io/f2mua/.

The following dataset was generated:

| Author(s) | Year | Dataset title | Dataset URL | Database and Identifier |
|---|---|---|---|---|
| Strube A, Rose M, Fazeli S, Büchel C | 2020 | The temporal and spectral characteristics of expectations and prediction errors in pain and thermoception | https://osf.io/f2mua/ | Open Science Framework, F2MUA |

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
