## [Decision Letter]

**Acceptance summary:**

Our expectations strongly influence how we perceive painful stimuli, as demonstrated by the well-known placebo effect. This paper elegantly elucidates the neural dynamics involved: expectations modulate low-frequency (alpha and beta) pre-stimulus oscillations, whereas expectation mismatches modulate post-stimulus high-frequency (gamma) power. This work demonstrates important overlap with expectation mechanisms in other perceptual domains (vision, audition), but also striking differences, such as that high-frequency power is decreased rather than increased by pain prediction errors.

**Decision letter after peer review:**

Thank you for submitting your article "The temporal characteristics of expectations and prediction errors in pain" for consideration by *eLife*. Your article has been reviewed by two peer reviewers, and the evaluation has been overseen by a Reviewing Editor and Floris de Lange as the Senior Editor. The following individual involved in review of your submission has agreed to reveal their identity: Enrico Schulz (Reviewer #1).

The reviewers have discussed the reviews with one another and the Reviewing Editor has drafted this decision to help you prepare a revised submission.

As the editors have judged that your manuscript is of interest, but as described below that additional analyses are required before it is published, we would like to draw your attention to changes in our revision policy that we have made in response to COVID-19 (https://elifesciences.org/articles/57162). First, because many researchers have temporarily lost access to the labs, we will give authors as much time as they need to submit revised manuscripts. We are also offering, if you choose, to post the manuscript to bioRxiv (if it is not already there) along with this decision letter and a formal designation that the manuscript is "in revision at *eLife*". Please let us know if you would like to pursue this option. (If your work is more suitable for medRxiv, you will need to post the preprint yourself, as the mechanisms for us to do so are still in development.)

Summary:

The study by Strube and colleagues investigates neural prediction error processing in the context of painful stimuli. The results suggest that alpha-to-beta oscillations were associated with pain expectation, whereas gamma band oscillations (GBO) were associated with prediction error (PE). The concept and the findings of the study are interesting. However, so far, it is not clear whether the findings unequivocally support the conclusions and additional analyses are needed.

Essential revisions:

1) There is a vast amount of literature on pain-related theta, alpha, and gamma activity that the authors should mention in the Introduction in order to prepare the reader for the results. Similarly, the Discussion section is very much focused on imaging effects despite a vast amount of literature on pain-related oscillations. Interestingly, the only paper the authors are citing for gamma activity is relying on an artefact (Zhang et al., 2012). The authors should discuss their findings in light of neurophysiological studies.

2) In contrast to Fazeli and Büchel, 2018, the behavioral data do not show PE effects. Does it make sense to analyze PE effects in the imaging data if no behavioural PE effects are found?

3) The study deals with expectations and PE in pain. To this end, a cueing paradigm has been performed with three thermal stimulus intensities. However, the lowest stimulus intensity has not been perceived as painful but as warm. Thus, some PE refer to errors regarding pain intensity and other PE to errors regarding pain vs non-painful warm. Moreover, the low pain cue is in fact a no pain cue. Does the study therefore really deal with PE in pain? Wouldn't it be necessary to restrict the analysis to painful stimuli? Or should the study be re-framed as studying PE in thermal perception?

Related to this, it is not clear whether it was a good decision to keep the stimulation temperature at the same level for all participants. As this is a within-subject design, the authors want to maximise the perceptual differences (e.g. no, low, high pain).

4) The current study conceptualizes PE as absolute unsigned PE, i.e., similar PE occur when the sensory evidence is less or more intense than expected. However, other PE concepts have assumed absolute signed PE, i.e., opposite neural effects occur when the sensory evidence is less or more intense than expected. Moreover, a third concept assumes negative PE, i.e., PE occur only when the sensory evidence is more intense than expected. The authors should clearly explain and motivate their PE definition. Moreover, the authors might perform similar analyses for other PE definitions.

5) The relationship between GBO and PE is negative, i.e., the greater the PE the lower the GBO amplitude. This pattern is at variance with all previous findings on GBO and PE and therefore contradicts rather than supports PE coding by GBO.

6) The relationship between alpha-to-beta-oscillations and expectations is not fully clear. The text and Figure 6 indicate a positive relationship whereas later it describes a decrease of alpha-to-beta band power associated with the manipulation of expectations, i.e., a negative relationship.

7) The expectation effects have been analyzed cue-locked. Thus, their timing with respect to thermal stimulus application is unclear, i.e. it is unknown whether expectations effects occur before or after stimulus application. The authors should find a way to clarify this.

8) Is there a reason why the authors focused on the pain part only? The visual part could serve as a control experiment, or the pain part as control for the visual experiment. It would be desirable to see whether there are similar effects of prediction error in either modalities. Alternatively, the authors might explain the motivation of the visual control condition and why it is not relevant for the present study.

9) There are problems with the analysis of gamma oscillations. There are additional and essential steps required in order to prevent that the findings are based on muscle artefacts (such as in Zhang et al., 2012). The authors may want to take the following steps:

a) Inspection of time-frequency decomposed ICA data at single trial level

b) Plot the trials after z-transforming based on the mean and the standard deviation of the entire component or potentially across all components. The z-transformation should be done separately for each frequency.

c) Artefact components or trials are easily visible and should be compared with the raw ICA time course as the muscle spikes can be easily detected there, too. However, the TFR plots are more sensitive. The artefact detection procedure may require a finer sliding window than the 50 ms which the authors used.

d) Single and separate muscle spikes are shown as columns, similar to the figure presented in Zhang (2012). Overlapping muscle spikes appear like "clouds" and can easily be misinterpreted as cortical activity. A sensitive single trial inspection on ICA transformed data is helpful.

e) As the authors have a low number of trials for some combinations (~1%), they may want to focus more on component rejection than on trial rejection. The authors may be required to remove up to 30 components from further analyses. The Vision Analyzer software from Brain Products has some features the authors may find useful, which is a Matlab interface for data export to FieldTrip as well as an excellent overlay function of cleaned vs uncleaned data after component removal.

10) Usually, less than half of the sample exhibits pain-related gamma activity. Could the authors provide a histogramme plot for the baseline-corrected gamma amplitude across the sample?

11) Is there a reason for not including a source analysis for pain intensity encoding? The authors provided a source analysis for all other aspects and should do the same for pain encoding for all frequencies.

12) The analysis should be explained in sufficient detail for replication. In particular, it should be explained for which time-frequency-electrode spaces cluster permutation tests were performed. Moreover, it should be detailed how prediction error effects were calculated (interactions between stimulus intensity and expectations in ANOVA?).

[Editors' note: further revisions were suggested prior to acceptance, as described below.]

Thank you for resubmitting your article "The temporal characteristics of expectations and prediction errors in pain" for consideration by *eLife*. Your revised article has been reviewed by two peer reviewers, and the evaluation has been overseen by a Reviewing Editor and Floris de Lange as the Senior Editor. The following individual involved in review of your submission has agreed to reveal their identity: Enrico Schulz (Reviewer #1).

Summary:

The reviewers felt that the revisions have improved the manuscript, and the paper has clear potential. However, substantial revisions are still required to make this paper suitable for publication. They point out that several central issues have not been fully addressed for far.

Essential Revisions:

1) One issue that not been fully address are the concerns about the pain rating scale, where the "1" is clearly outside the pain range, being rated as neutral. Moreover, the low pain cue is in fact a no pain cue. What matters is not located in the periphery and the use of C-fibres, but the subjective pain experience of the participants. There's plenty of literature showing that participants exhibit differences in pain sensitivity. The QUEST algorithm to individually adapt pain intensities from Taesler and Rose, 2017, would have been more suitable to define different levels of pain intensity. Within-subject analyses should (probably) always use individually adapted pain intensities. This issue that the stimuli were ranging from non-painful heat to moderate pain needs to be explicitly addressed throughout the manuscript (including the title) and added to the paragraph on limitations. It is essential that the following questions are addressed convincingly and that these considerations are included to the Discussion section. Does the study really deal with PE in pain? Wouldn't it be necessary to restrict the analysis to painful stimuli? Or should the study be re-framed as studying PE in thermal perception?

2) The answer to the central question about the absence of behavioural effects is not fully convincing. The main argument is simply that it does make sense to investigate PE imaging effects without any behavioural effects. However, a more substantiated consideration of the significance of PE imaging effects in the absence of behavioural effects would be appropriate. These considerations should also be included in the Discussion section.

3) The relationship between GBO and PE is negative, i.e., the greater the PE the lower the GBO amplitude. This pattern is at variance with all previous findings on GBO and PE and therefore contradicts rather than supports PE coding by GBO. The authors reply to this point in the previous round was not sufficiently clear. Clear arguments in plain terms are needed. Moreover, the unusual inverse coding of PEs by gamma oscillations should be added to the Abstract.

4) The cluster-based permutation test is central to control for false positives. It has now been clarified that clustering has been performed across electrodes. However, it is best practice to cluster across electrodes, time and space. It is therefore essential to adjust the clustering procedure.

5) It is unclear whether it is appropriate to apply a baseline correction in reference to the entire trial segment. Could the authors provide a reference in order to justify their approach? Otherwise, they might consider applying a conventional baseline correction.

6) The reviewers suggest the authors consider dropping the source localisation. Most of the images do not seem to make much sense. The only exception that resembles a meaningful solution for the expectation analysis appears to be exaggerated. The authors rightfully mention in the manuscript that the anterior insula is involved in cognitive processing, such as expectation. However, the activity cluster points to the posterior insula and has its major part extended to the lingual gyrus. It does not seem appropriate to rely on the interpretation of one (out of many) remotely interpretable source analysis. It suggests selectively interpreting results that fit the hypotheses.

7) For the interpretation of the EXP and PE effect the authors should be sure that there are no differences in subjective pain intensity between the 3 conditions of EXP and the 2 conditions of PE. The question is whether the prediction error is a real error or whether the pain stimuli were "naturally" experienced as more or less painful than intended. Previous studies have shown that the trial-by-trial experience of pain can substantially jitter within subjects, even without any kind of intervention.

8) It is not clear why there is a significant cluster for expectation in the alpha/beta range between 1 – 2 s after cue onset. From visual inspection and "plausibility check" this cluster does not particularly stick out from the scattered apparently insignificant clusters across the entire time-frequency range. The F-values at 2.5s/70Hz even appear to be higher than the significant cluster.

Reviewer #1:

The EEG study by Strube and colleagues used a predictive model paradigm to investigate the temporal dynamics of expectation and prediction errors on the processing of heat and pain. Their elaborate and balanced paradigm allowed to differentiate the neuronal oscillations contributing to the encoding of (a) stimulus intensity, (b) expectation, and (c) prediction errors. As a major weakness of the study, the lowest stimulus intensity was not perceived as painful, which does not justify restricting the interpretation to the pain domain but must include heat processing. The reason for this disadvantage is that the stimulus intensity has not been adapted to the participants. All study participants received the same stimulus intensities. This adaptation is mandatory due to the large differences in pain sensitivity across individuals and the focus on within-subject statistical analyses.

The study corroborates previous work on the influence of cognitive factors on the processing of pain. Furthermore, the study also takes advantage of the higher temporal resolution of EEG, which enabled the authors to analyse the data in reference to the onset of the expectation phase, as well as to analyse the data in reference to the subsequent (jittered) onset of the pain perception phase. As a result, the authors associated pain encoding with gamma activity and high-alpha/low-beta activity for predictor error encoding. The authors utilised an established design, which they have already published using fMRI.

The study is of utmost relevance for a broad readership of scientists in the fields of pain and cognition. The authors applied a timely randomisation algorithm at cluster-level in order to correct for multiple testing. They also applied an EEG source localisation of their effects with barely interpretable results. The many presented source maps are a good example for the challenges of EEG source localisation, which probably often do not exhibit results we can rely on. This can cause severe publication bias, where "good" results are published and "bad" results are dropped. The source localisation could have been improved by the use of individual and accurate electrode positions (instead of standard electrode positions), with the use of individual 3D brain images to account for individual anatomical differences, as well as by the co-registration of EEG electrode positions to the individual head shape.

Reviewer #2:

The revisions have significantly improved the manuscript. Many details have been clarified. However, some important issues should be addressed in more detail.

In contrast to Fazeli and Büchel, 2018, the behavioral data do not show PE effects. Does it make sense to analyze PE effects in the imaging data if no behavioral PE effects are found?

The study deals with expectations and PE in pain. To this end, a cueing paradigm has been performed with three thermal stimulus intensities. However, the lowest stimulus intensity has not been perceived as painful but as warm. Thus, some PE refer to errors regarding pain intensity and other PE to errors regarding pain vs non-painful warm. Moreover, the low pain cue is in fact a no pain cue. Does the study therefore really deal with PE in pain? Wouldn't it be necessary to restrict the analysis to painful stimuli? Or should the study be re-framed as studying PE in thermal perception?

The relationship between GBO and PE is negative, i.e., the greater the PE the lower the GBO amplitude. This pattern is at variance with all previous findings on GBO and PE and therefore contradicts rather than supports PE coding by GBO.

The analysis should be explained in sufficient detail for replication. In particular, it should be explained for which time-frequency-electrode spaces cluster permutation tests were performed. Moreover, it should be detailed how prediction error effects were calculated (interactions between stimulus intensity and expectations in ANOVA?).

---

## [Author Response]

Essential revisions:1) There is a vast amount of literature on pain-related theta, alpha, and gamma activity that the authors should mention in the Introduction in order to prepare the reader for the results. Similarly, the Discussion section is very much focused on imaging effects despite a vast amount of literature on pain-related oscillations. Interestingly, the only paper the authors are citing for gamma activity is relying on an artefact (Zhang et al., 2012). The authors should discuss their findings in light of neurophysiological studies.

Thank you for pointing this out. We have now integrated several paragraphs in the Introduction as well as in the Discussion part of the revised paper and added references to existing ones, focusing more on neurophysiological studies of pain.

2) In contrast to Fazeli and Büchel, 2018, the behavioral data do not show PE effects. Does it make sense to analyze PE effects in the imaging data if no behavioural PE effects are found?

This is an important question which pertains to neuroscientific studies in general. In essence, an experiment links a perturbation (in our case expectations, heat stimulation and prediction errors) to a readout (behavioral response, neurophysiological response, autonomic response etc.). In many cases a behavioral readout is essential, because this is the effect that determines whether the perturbation is effective, e.g. in many clinical studies, where a successful perturbation has to change disease symptoms (e.g. pain as indicated by pain ratings). However, in studies trying to understand neurophysiological mechanisms it is meaningful to interpret a neurophysiological readout as a result of a perturbation even in the absence of a behavioral effect. In addition, a VAS rating (or any behavioral read-out) integrates many neural processes and although careful experimental design tries to isolate the effects of interest, it is still subject to many cognitive and emotional processes which are not directly involved in the particular process in question (e.g. a prediction error). Consequently, the behavioral response is affected by many (unrelated) influences, which can obfuscate the effect in question.

In the study by Geuter, Boll, Eippert and Büchel, 2017, we observed a prediction error effect using autonomic measures (SCR, pupil). However, in the study using a more complex paradigm (Fazeli and Büchel, 2018) where we used pain ratings, we were not able to reveal a behavioral prediction error effect (similar to the present study). It is therefore possible, that behavioral pain ratings are less sensitive to PEs and that it would have been a better choice to have used autonomic readouts.

3) The study deals with expectations and PE in pain. To this end, a cueing paradigm has been performed with three thermal stimulus intensities. However, the lowest stimulus intensity has not been perceived as painful but as warm. Thus, some PE refer to errors regarding pain intensity and other PE to errors regarding pain vs non-painful warm. Moreover, the low pain cue is in fact a no pain cue. Does the study therefore really deal with PE in pain? Wouldn't it be necessary to restrict the analysis to painful stimuli? Or should the study be re-framed as studying PE in thermal perception?Related to this, it is not clear whether it was a good decision to keep the stimulation temperature at the same level for all participants. As this is a within-subject design, the authors want to maximise the perceptual differences (e.g. no, low, high pain).

Heat sensitive C-fiber nociceptors have a median heat threshold of 41°C (Treede et al., 1995) and therefore our lowest stimulus temperature (42°C ) was set above this threshold. Although this stimulus intensity was not always rated as painful (see below), we still believe that this choice is valid, based on the neurophysiological properties of C fibers. As a consequence, it is valid to assume that these stimuli activate the nociceptive system and thus can cause pain. In addition, we used a rather coarse VAS scale with only four levels to allow brief button presses, which might have introduced rating ambiguities at the lower pain levels. Finally, using three temperatures allowed us to cover a large range of temperatures associated with nociception. We also wanted to stay as close to a previous fMRI study as possible and use the same stimulus properties (Fazeli and Buechel 2018). This has now been clarified in the manuscript.

4) The current study conceptualizes PE as absolute unsigned PE, i.e., similar PE occur when the sensory evidence is less or more intense than expected. However, other PE concepts have assumed absolute signed PE, i.e., opposite neural effects occur when the sensory evidence is less or more intense than expected. Moreover, a third concept assumes negative PE, i.e., PE occur only when the sensory evidence is more intense than expected. The authors should clearly explain and motivate their PE definition. Moreover, the authors might perform similar analyses for other PE definitions.

Thank you for this suggestion. We have extended our analysis and have explored additional prediction errors. Initially we refrained from such an analysis, because a signed PE is not orthogonal to the expectation factor. However, assuming that an expectation should be observed early (after the cue) and a PE later (after the nociceptive stimulus), it is possible to test for a signed PE at the pain phase. We have now performed these analyses. However, they did not reveal any significant effect (all p>0.05). This has now been added to the manuscript. Additionally, we investigated a “negative PE” that we refer to as “Pain PE” (as in Geuter et al., 2017). Essentially, this resulted in a factor where only 3 cue-stimulus combinations resulted in a PE, namely: Low Pain Cue + Medium Pain / Low Pain Cue + High Pain / Medium Pain Cue + High Pain. However, this did not reveal any significant results (all p >.05).

5) The relationship between GBO and PE is negative, i.e., the greater the PE the lower the GBO amplitude. This pattern is at variance with all previous findings on GBO and PE and therefore contradicts rather than supports PE coding by GBO.

We included two lines of thought in the Discussion to relate these results to the literature associating PEs with GBO. Firstly, pre-stimulus γ oscillations have been found to modulate the subsequent pain response negatively, i.e. pain was perceived as more painful and evoked potentials showed a larger amplitude with a lower pre-stimulus γ activity (Tu et al., 2016), which implies a comparable mechanisms to our prediction error finding. Secondly, the γ response related to stimulus intensity without a prediction error entails an increase of GBO (Figure 5), which differs for the different stimulus intensities and has to be taken into account. We have now added a paragraph in the Discussion and describe this notion:

“However, from a different point of view, an early negative modulation of γ activity is directly associated with a relative γ increase over time (see Figure 7—figure supplement 3 for a schematic demonstration), suggesting a potential mechanism of the neuronal encoding of prediction errors in pain. In addition, pre-stimulus gamma oscillations modulate the subsequent pain response negatively, i.e. pain was perceived as more painful and evoked potentials showed a larger amplitude with a lower pre-stimulus gamma activity (Tu et al., 2016).”

6) The relationship between alpha-to-beta-oscillations and expectations is not fully clear. The text and Figure 6 indicate a positive relationship whereas later it describes a decrease of alpha-to-beta band power associated with the manipulation of expectations, i.e., a negative relationship.

We apologize for this mistake in the manuscript, which has now been corrected. We now correctly refer to the POSITIVE relationship in all parts of the manuscript.

7) The expectation effects have been analyzed cue-locked. Thus, their timing with respect to thermal stimulus application is unclear, i.e. it is unknown whether expectations effects occur before or after stimulus application. The authors should find a way to clarify this.

Thank you for pointing this out. With our revised analysis, the expectation effects expands clearly to time points before stimulus onset. To clarify this in the revised manuscript, we marked the area of the stimulus onset in the time-frequency representation of the effect.

8) Is there a reason why the authors focused on the pain part only? The visual part could serve as a control experiment, or the pain part as control for the visual experiment. It would be desirable to see whether there are similar effects of prediction error in either modalities. Alternatively, the authors might explain the motivation of the visual control condition and why it is not relevant for the present study.

It was our primary aim to identify effects with high temporal resolution using EEG and at the same time stay as close as possible to our previous fMRI study (Fazeli and Büchel, 2018). Therefore, we used an experimental design that was almost identical to this previous study. We agree with this reviewer, that the visual part of the experiment is interesting in its own right and we plan to jointly analyze the EEG data from this study together with the fMRI data from Fazeli and Büchel, 2018 and summarize the results in another paper. We strongly believe that this would be beyond the scope of the current manuscript with a focus on the investigation of the properties of nociceptive responses.

9) There are problems with the analysis of gamma oscillations. There are additional and essential steps required in order to prevent that the findings are based on muscle artefacts (such as in Zhang et al., 2012). The authors may want to take the following steps:a) Inspection of time-frequency decomposed ICA data at single trial levelb) Plot the trials after z-transforming based on the mean and the standard deviation of the entire component or potentially across all components. The z-transformation should be done separately for each frequency.c) Artefact components or trials are easily visible and should be compared with the raw ICA time course as the muscle spikes can be easily detected there, too. However, the TFR plots are more sensitive. The artefact detection procedure may require a finer sliding window than the 50 ms which the authors used.d) Single and separate muscle spikes are shown as columns, similar to the figure presented in Zhang et al., 2012. Overlapping muscle spikes appear like "clouds" and can easily be misinterpreted as cortical activity. A sensitive single trial inspection on ICA transformed data is helpful.e). As the authors have a low number of trials for some combinations (~1%), they may want to focus more on component rejection than on trial rejection. The authors may be required to remove up to 30 components from further analyses. The Vision Analyzer software from Brain Products has some features the authors may find useful, which is a Matlab interface for data export to FieldTrip as well as an excellent overlay function of cleaned vs uncleaned data after component removal.

This is a major point and we thank the reviewer for pointing this out and in addition giving us clear guidance in the form of a step-by-step approach for the removal of artifacts. We have now reanalyzed our entire data set following this approach. The revised manuscript now includes a detailed description of these steps (which led to up to 30 rejected components) and a histogram showing the number of rejected artifacts per participant for an overview (Figure 1—figure supplement 1). Importantly, we were able to replicate all effects using this more rigorous approach and have changed the results and figures in the revised manuscript accordingly.

10) Usually, less than half of the sample exhibits pain-related gamma activity. Could the authors provide a histogramme plot for the baseline-corrected gamma amplitude across the sample?

Thank you for pointing this out. We have now provided a histogram for the baseline-corrected gamma amplitude for low and high pain conditions. Moreover, we added a plot showing the difference between low and high pain conditions (Figure 5—figure supplement 4).

11) Is there a reason for not including a source analysis for pain intensity encoding? The authors provided a source analysis for all other aspects and should do the same for pain encoding for all frequencies.

We have now included a source analysis for all neurophysiological effects in the supplementary materials.

12) The analysis should be explained in sufficient detail for replication. In particular, it should be explained for which time-frequency-electrode spaces cluster permutation tests were performed. Moreover, it should be detailed how prediction error effects were calculated (interactions between stimulus intensity and expectations in ANOVA?).

Thank you for pointing this out, we have now expanded on our explanations regarding our analysis, especially for the time-frequency-electrode spaces as well as for the calculation of the absolute prediction error effect.

[Editors' note: further revisions were suggested prior to acceptance, as described below.]

Essential Revisions:1) One issue that not been fully address are the concerns about the pain rating scale, where the "1" is clearly outside the pain range, being rated as neutral. Moreover, the low pain cue is in fact a no pain cue. What matters is not located in the periphery and the use of C-fibres, but the subjective pain experience of the participants. There's plenty of literature showing that participants exhibit differences in pain sensitivity. The QUEST algorithm to individually adapt pain intensities from Taesler and Rose, 2017, would have been more suitable to define different levels of pain intensity. Within-subject analyses should (probably) always use individually adapted pain intensities. This issue that the stimuli were ranging from non-painful heat to moderate pain needs to be explicitly addressed throughout the manuscript (including the title) and added to the paragraph on limitations. It is essential that the following questions are addressed convincingly and that these considerations are included to the Discussion section. Does the study really deal with PE in pain? Wouldn't it be necessary to restrict the analysis to painful stimuli? Or should the study be re-framed as studying PE in thermal perception?

Although our lowest stimulus temperature was above the threshold for peripheral nociceptors, we agree that at least some of the low pain trials did not lead a subjective pain experience. Therefore, we have followed the suggestion by the reviewers and performed an additional analysis, which only included the medium and high intensity heat stimuli. The behavioral data analysis shows clear effects of stimulus intensity, expectation and prediction error (PE) effect. This is similar to the analysis comprising all stimuli, yet the PE effect, which was a trend (p<0.1) in the complete analysis is now significant at p<0.001. Consistent with the original analysis, the new EEG analysis with only the medium and high stimulus intensities, revealed a negative absolute prediction error effect in the gamma band and an alpha-to-beta effect of stimulus intensity. These results have now been incorporated into the manuscript including a new figure.

2) The answer to the central question about the absence of behavioural effects is not fully convincing. The main argument is simply that it does make sense to investigate PE imaging effects without any behavioural effects. However, a more substantiated consideration of the significance of PE imaging effects in the absence of behavioural effects would be appropriate. These considerations should also be included in the Discussion section.

While reducing the design to medium and high intensity stimuli resulted in a significant prediction error effect in the behavioral data, we included an elaborate and more general discussion of whether it makes sense to investigate PE imaging effects without any behavioral effects based on the trend level effect of the full model.

“We observed a behavioral effect of prediction errors on perceived stimulus intensity in the reduced pain model, but this effect was only a trend in the full model. The latter finding replicates a previous study (Fazeli and Büchel, 2018) indicating a robust effect. Interestingly, the effect of prediction errors on perception increased, and became significant, when we constrained our analysis to the clearly painful stimuli (reduced pain model). This suggests that a prediction error seems to more strongly affect pain perception whereas the effect is weaker in the context of thermoception. However, this speculation should be corroborated in a future study.

On a more conceptual level, the investigation of neurophysiological effects even in the absence of a behavioral effect has been considered meaningful (Wilkinson and Halligan, 2004). In particular, the authors argue that because it is commonly unknown which parts of a cognitive process (and in which way) produce a specific behavioral response the relationship between neurophysiological data and behavioral responses should not be overemphasized, and therefore it can be misleading to declare behavioral effects a reference or “gold standard”. Studies aiming to understand neurophysiological mechanisms of cognition, usually relate a neurophysiological readout to a known perturbation (i.e. experimental design), which is meaningful in its own right.”

3) The relationship between GBO and PE is negative, i.e., the greater the PE the lower the GBO amplitude. This pattern is at variance with all previous findings on GBO and PE and therefore contradicts rather than supports PE coding by GBO. The authors reply to this point in the previous round was not sufficiently clear. Clear arguments in plain terms are needed. Moreover, the unusual inverse coding of PEs by gamma oscillations should be added to the Abstract.

Thank you for this comment. We now explicitly state that our findings contradict a gamma response by prediction errors in the manuscript (i.e. Abstract and Discussion). However, we now also discuss that our findings are in line with the direction of gamma responses in word matching and working memory experiments. In this literature, successful matching (i.e. the absence of a prediction error) between external input and internal representation induces gamma power increases (Herrmann et al., 2004; Osipova et al., 2006; Wang et al., 2018), which is in line with our findings.

4) The cluster-based permutation test is central to control for false positives. It has now been clarified that clustering has been performed across electrodes. However, it is best practice to cluster across electrodes, time and space. It is therefore essential to adjust the clustering procedure.

We apologize that we have not been clearer in our previous revision. We have now clarified that clustering was indeed performed across electrodes, time and space: “Samples (exceeding the threshold of F(1,28) = 4.196) were clustered in connected sets on the basis of temporal (i.e. adjacent time points), spatial (i.e. neighboring electrodes) and spectral (i.e. +/- 1Hz) adjacency.”

5) It is unclear whether it is appropriate to apply a baseline correction in reference to the entire trial segment. Could the authors provide a reference in order to justify their approach? Otherwise, they might consider applying a conventional baseline correction.

Although this approach is not unusual (Grandcamp and Delorme, 2011), we followed the reviewers’ advice and re-analyzed our data with a conventional baseline approach and changed the results part accordingly. We thank the reviewers for the suggestion to change the baseline correction approach as this clearly improved data quality and sensitivity. We describe our baseline approach as follows:

“Spectral estimates were averaged for each subject over trials. Afterwards, a z-baseline normalization was performed based on a 500ms baseline before cue onset. For cue-locked data, a time frame ranging from -650ms to -150ms was chosen as a baseline. A distance from the cue onset to the baseline period of 150ms was set because of the half-taper window length of 150ms, i.e. data points between -150ms and 0ms are contaminated by the onset of the cue. For stimulus-locked trials, a variable cue duration (1500-1900ms) and a variable stimulus offset based on the ramp-up time (225-375ms) were additionally taken into account, resulting in an according baseline from -2950ms to -2450ms from pain onset. For the baseline correction of time-frequency data, the mean and standard deviation were calculated for the baseline period (for each subject-channel-frequency combination, separately). The mean spectral estimate of the baseline was subtracted from each data point, and the resulting baseline-centered values were divided by the baseline standard deviation (classical baseline normalization – additive model; see Grandchamp and Delorme, 2011).”

6) The reviewers suggest the authors consider dropping the source localisation. Most of the images do not seem to make much sense. The only exception that resembles a meaningful solution for the expectation analysis appears to be exaggerated. The authors rightfully mention in the manuscript that the anterior insula is involved in cognitive processing, such as expectation. However, the activity cluster points to the posterior insula and has its major part extended to the lingual gyrus. It does not seem appropriate to rely on the interpretation of one (out of many) remotely interpretable source analysis. It suggests selectively interpreting results that fit the hypotheses.

We have dropped the source localization as suggested.

7) For the interpretation of the EXP and PE effect the authors should be sure that there are no differences in subjective pain intensity between the 3 conditions of EXP and the 2 conditions of PE. The question is whether the prediction error is a real error or whether the pain stimuli were "naturally" experienced as more or less painful than intended. Previous studies have shown that the trial-by-trial experience of pain can substantially jitter within subjects, even without any kind of intervention.

We are not entirely sure if we fully understand this comment. We guess that the statement “…the authors should be sure that there are no differences in subjective pain intensity between the 3 conditions of EXP and the 2 conditions of PE…” contains a typo, as (i) the full design elicits 3 levels of expectations and 3 level of prediction errors and (ii) differences in subjective pain due to expectation and prediction errors are the focus of the paper.

Nevertheless, we have several ideas what could be meant here:

1) The three factors (INT, EXP and PE) are orthogonal (uncorrelated), hence differences in Stimulus Intensity do not affect factor estimates of EXP and PE.

2) The largest part of the expectation cluster is associated with activity preceding the heat stimulus, and thus the expectation effect is not affected by differences associated with subjective pain.

3) We also performed the behavioral data analysis only on trials which remained in the EEG data analysis after trial rejection. This did not change the results, as all comparisons remained significant.

4) Trial-by-trial differences of pain perception with identical stimulus intensities are indeed very common. However, these differences cannot systematically bias our results as they “end up” in the residuals of our statistical model.

8) It is not clear why there is a significant cluster for expectation in the alpha/beta range between 1 – 2 s after cue onset. From visual inspection and "plausibility check" this cluster does not particularly stick out from the scattered apparently insignificant clusters across the entire time-frequency range. The F-values at 2.5s/70Hz even appear to be higher than the significant cluster.

While in cluster permutation testing large clusters with lower F-values can reach a higher cluster mass than clusters with apparently higher F-values at single samples, the new analysis with a different baseline approach now resulted in a very clear low frequency alpha-to-beta cluster for the expectation effect.